# cgDDI: Controllable Generation of Diverse Dermatological Imagery for Fair and Efficient Malignancy Classification

## Abstract

Skin diseases impact the lives of millions of people around the world from different backgrounds and ethnicities. Therefore, accurate diagnosis in the dermatological domain requires focused work toward fairness in different skin-toned populations. However, a significant lack of expertly annotated dermatological images, especially those describing underrepresented skin tones and rare diseases, slows progress toward broadly accurate models and clear fairness metrics. In this work, we introduce **C**ontrollable **G**eneration of **D**iverse **D**ermatological **I**magery (**cgDDI**), a method capable of (1) synthesizing pixel-perfect in-distribution healthy samples, (2) lesion-mapping extremely rare lesions onto novel skin-tone combinations without training and (3) efficient high-fidelity parametric generation with as few as 10 training samples. Our approach is controllable via learned disease-specific prompts or skin tone descriptors, either visually or textually, allowing for selection of key sensitive attributes. We leverage cgDDI to grow a 656 real-image dataset by more than $400\times$. The resulting skin-tone-balanced dataset enables the development of accurate classification systems along with significant improvement on essential fairness metrics. Malignancy classification experiments on the Diverse Dermatology Images (DDI) benchmark shows our method reaches competitive performance (86.4% accuracy) when trained exclusively on our synthetic data and state-of-the-art performance (90.9% accuracy) when fine-tuned on real data. Additionally, we achieve leading metrics for Predictive Quality Disparity, Demographic Disparity, Equality of Opportunity as well as equitable generative image quality measurements for underrepresented skin-tones and rare diseases. We publish code, model weights, and generated datasets at `https://anonymous.4open.science/r/ControllableGenDDI` in support of further research in this direction.

## 1 Introduction

Achieving fairness in medical artificial intelligence (AI) requires addressing fundamental challenges in data generation for domains with severe data scarcity and demographic imbalances. Dermatological AI exemplifies these challenges: existing datasets suffer from limited expert annotations, poor representation of darker skin tones, and extreme rarity of certain conditions (Daneshjou et al., 2021). While generative models offer a potential solution, current approaches require large training sets (Akrout et al., 2024; Ktena et al., 2024), do not cover the full skin-tone spectrum (Wang et al., 2024), or ignore extremely rare diseases (Sagers et al., 2022). We present a novel hybrid generation framework that addresses these limitations through complementary parametric and non-parametric approaches, achieving data-efficient synthesis while maintaining fairness across populations.

Skin diseases affect millions globally, with expert diagnosis accuracy being 4% lower for darker-skinned patients (Groh et al., 2024) and 3 billion people lacking adequate dermatological care (Coustasse et al., 2019). Early detection of skin cancers significantly increases survival rates (Balch et al., 2009), yet the average wait time exceeds 38 days in the United States alone (Tsang & Resneck, 2006). While AI systems have shown promise in skin lesion classification (Winkler et al., 2023; Esteva et al., 2017), a survey of 70 dermatological AI studies found less than 25% included ethnicity and only 10% included skin-tone descriptors (Daneshjou et al., 2021). These disparities stem from technical challenges our method directly addresses:

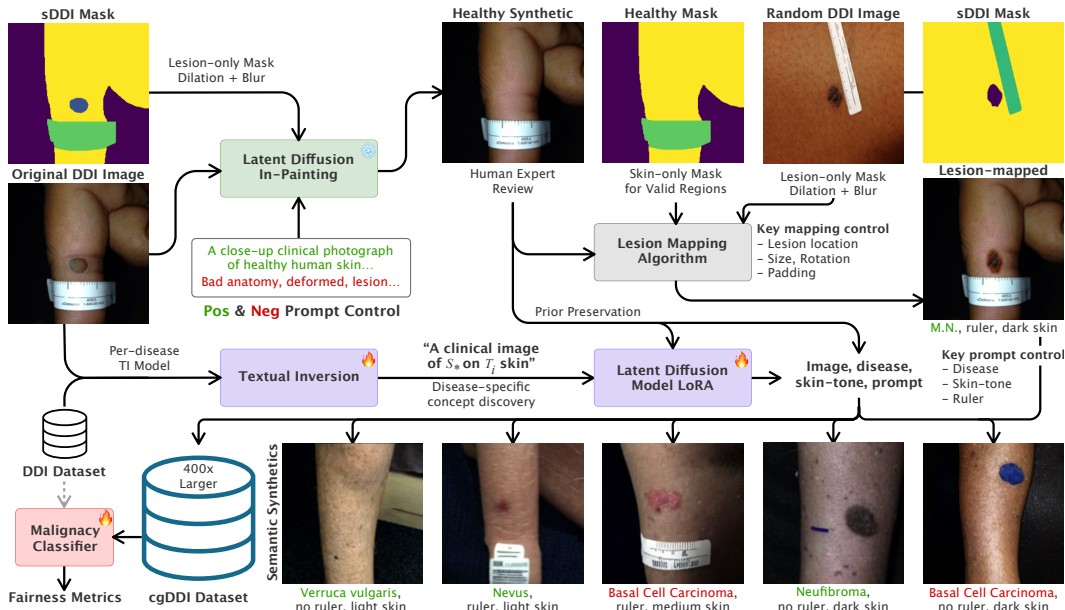

Figure 1: **cgDDI framework**. We generate dermatological synthetics in a controllable manner. First original images, masks and prompts create **healthy synthetics**. These are used to create **lesion-mapped synthetics** from real image donors, as prior preservation, and as semantic prompts. Additionally, we learn disease-specific concepts to train a latent diffusion model from which **semantic synthetics** are sampled. Finally, cgDDI is aggregated and used to learn **fair classification** networks. Abbreviation: "M.N." shortens Melanocytic Nevi.

- **Data Scarcity:** Expertly annotated medical imagery is often limited and costly to acquire due to privacy concerns, the rarity of the disease, and the availability of medical professionals and some prior work retains these datasets for internal use only (Akrout et al., 2024; Ktena et al., 2024).

- **Uneven Distribution:** The real-world distribution of various skin lesions is naturally imbalanced. This disparity is further complicated when considering skin-tone balance, especially since existing datasets are predominantly collected from light-skinned populations (Groh et al., 2021).

- **Morphology and Demographics:** The morphology of the disease and general appearance can differ between individuals due to factors such as ethnicity, sex, or age (Adelekun et al., 2020). Skin lesions naturally manifest in different areas of the body (e.g., face, back, leg, etc.), making the collection of real-world datasets covering the full distribution difficult.

Simultaneously, the size of the dataset is not the only important dimension to consider when addressing scarcity; a dataset may be large but lack sufficient population, lesion, and morphological diversity, potentially leading to poor generalization (Daneshjou et al., 2022b). These factors limit the ability of the research community to develop fair and generalized models. We present a novel hybrid generation framework that addresses fundamental challenges in synthesizing fair and diverse medical imagery under extreme data constraints. Our key methodological contributions include:

- **cgDDI Framework:** A method capable of dermatological image generation in a controllable manner, allowing for selection of key sensitive attributes (e.g., disease, skin-tone, etc.) or conditioned on input images for nuanced semantic control (e.g. body-part, markings, ruler, etc.). Single-sample observations are effectively augmented non-parametrically via lesion-mapping while more common cases are efficiently learned using prior preservation loss, leveraging healthy samples as regularization. The framework is visualized in Figure 1.

- **cgDDI Dataset:** We synthesize a dataset containing three distinct data types, namely: healthy (lesion-less), lesion-mapped (from real donor images), and semantic (conditioned on learned disease and skin tone features) synthetics. Each sample is accompanied by

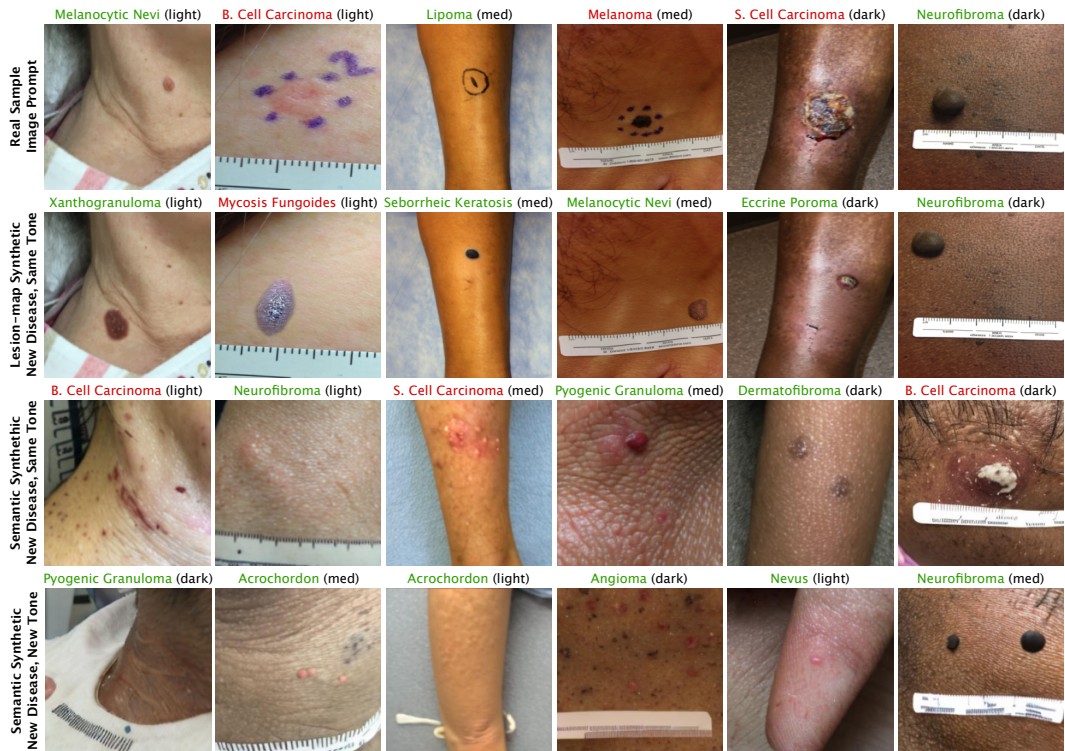

Figure 2: **cgDDI Dataset**. The first row shows real samples used for prompting. The second row contains a novel lesion from a donor sample transplanted onto the prompt image. Third and fourth rows contain sampled synthetics, conditioned on the prompt image, a novel target disease (malignant or benign), and a target skin-tone (light, medium, or dark). Abbreviations: "S." shortens "Squamous", "B." shortens "Basal".

rich metadata including full captions, generation parameters, skin tone, condition, and malignancy. Our dataset is balanced across populations by growing a dataset of 656 real image bases by more than $400\times$. A sample of cgDDI images are visualized on Figure 2.

- **Classification and Generative Fairness:** We measure the diagnostic quality of our data by training a malignancy classifier and evaluating various fairness metrics against real data. When training purely on our synthetic data, we achieve the competitive performance of 86.4% overall accuracy. After fine-tuning on a small portion of real data, we achieve leading performance at 90.9% overall accuracy. In both training settings, fairness metrics improve over prior work and ablate generative fairness by method and skin-tone.

- **Open Datasets, Models, and Code:** We fully release all training code, models, and datasets containing 266,136 new synthetic images of three types, alongside metadata including skin tone, disease, textual captions, and other descriptors to encourage further fairness research.

## 2 RELATED WORK

**Real Datasets**  There are two main branches of dermatological datasets: dermoscopic (Tschandl, 2018; Hernández-Pérez et al., 2024), which requires a dermatoscope magnifier, and macroscopic (Groh et al., 2021; Daneshjou et al., 2022b), which are more similar to naked eye observations. We focus our work on the latter domain, as it is more likely the type of imaging encountered in first-pass clinical practice in low-resource environments. Further, we consider datasets that contain skin tone descriptors, as these descriptors are necessary for fairness evaluation.

The macroscopic dataset most commonly cited when training and evaluating fairness in dermatological AI is Fitzpatrick17k (F17k), primarily due to its relatively large size of around 17,000

Table 1: **Synthetic dermatological datasets.**

| Method | Training Data | Fairness Metrics | FST Coverage | Controllable $n$ Diseases | Total Size | Dataset Availability |
|---|---|---|---|---|---|---|
| Sagers et al. (2022) | F17k | FST Acc. | I-VI | 3 | 192 | Private |
| Sagers et al. (2023) | F17k, DDI | FST Acc. | I-VI | 9 | 459k | Published |
| Akrout et al. (2024) | Private | None | None | 6 | 180k | Private |
| Ktena et al. (2024) | Private* | FST Gap | I-VI | 27 | 50k | Private |
| Wang et al. (2024) | F17k | FST Acc. | I-II, V-VI | 7 | 7.6k | Private |
| **cgDDI (ours)** | DDI | Multiple | I-VI | 13 | 266k | Published |

*We note Ktena et al. (2024) training data can be made available for non-commercial use upon request, payment of administrative fees, and legal compliance.

samples. However, it suffers from skin tone imbalance, with 3.6 times more light than dark-skinned samples, and is annotated mainly by non-experts, leading to over 30% disease-label noise (Groh et al., 2021). Moreover, conditions determined without biopsy and histopathological evidence might lead to unreliable ground truth (Daneshjou et al., 2022a). Recent crowd-sourced datasets such as SCIN (Jeong et al., 2024), while balanced, share this limitation. The DDI dataset (Daneshjou et al., 2022b), on the other hand, is fully biopsy-confirmed and verified by two board-certified dermatologists. It provides supervised Fitzpatrick-scale (FST) (Fitzpatrick, 1988) scores for skin tones in all images, categorizing skin tones into light (I-II), medium (III-IV), or dark (V-VI), and is relatively balanced in skin tone. In addition, sDDI (Carrión & Norouzi, 2023) contributes human-annotated segmentation masks for a subset of DDI images. However, the primary limitations of DDI are its small size of 656 total samples (334 of which have mask annotations in sDDI) and naturally-occurring disease imbalance.

**Generative Approaches**    To address the challenges of small medical datasets, generative methods have been explored. Specifically, the success of Generative Adversarial Networks (GANs) (Goodfellow et al., 2020) and Diffusion Models (DMs) (Dhariwal & Nichol, 2021) has led to their use in the synthesis of photorealistic dermatology imagery (Wang et al., 2024; Akrout et al., 2024). Controllability is a key aspect in these generation tasks as it allows for the reconstruction of sensitive attributes (e.g., disease, skin tone, etc.), which is an area where GANs have struggled (Qin et al., 2020). DMs, on the other hand, pre-trained for image generation conditioned with textual information, have been shown to be controllable in the dermatological domain (Sagers et al., 2023), with textual inversion (Gal et al., 2023) leading to increased controllability (Wang et al., 2024). Yet, these frameworks have not been shown to be effective across the full range of skin tones or under a large number of disease conditions with various frequencies. DMs have also dominated the inpainting task (Razzhigaev et al., 2023), leading to inpainting and out-painting of dermatological data (Sagers et al., 2023).

**Synthetic Datasets**    We survey recent datasets created by controllable DMs and summarize their attributes in Table 1. Some related works do not use public training data (Akrout et al., 2024), and many do not publish their generated data (Sagers et al., 2022; Akrout et al., 2024; Ktena et al., 2024; Wang et al., 2024). Two have incomplete coverage of the skin-tone range (Akrout et al., 2024; Wang et al., 2024), and all collect only basic fairness metrics (per skin tone accuracy or performance gap). We note that training on F17k (Groh et al., 2021) can lead to a seemingly high visual quality due to its larger size, but diseases and skin tone representations could be noisy due to unreliable ground truth as cited earlier. Each approach has explicitly learned to generate a distinct number of conditions, often determined by the availability of training data.

**Classification Methods and Fairness**    Dermatological classification frameworks, previously studied using Convolutional Neural Network (CNNs) (Han et al., 2020), have adopted Vision Transformer (ViT) (Dosovitskiy et al., 2020) or DMs based approaches (Carrión & Norouzi, 2023), leading to increased performance. One commonality between ViTs, Large Vision Models (LVMs), or high-fidelity generative methods is that they often require large amounts of data to train effectively (Moon et al., 2022), a common limitation in the medical fairness domain. Recent studies have established the state-of-the-art malignancy classification under DDI both in terms of accuracy and fairness by

leveraging contrastive disentanglement (Du et al., 2023) and patch alignment (Aayushman et al., 2024) for efficient learning. Critically, these studies include a collection of informative fairness metrics. We build upon these recent works by training from scratch purely on our synthetic data, then fine-tuning on real data and computing the same metrics, which we discuss in the following sections.

## 3   CGDDI FRAMEWORK AND DATA

Our framework generates three complementary types of synthetic dermatological imagery through a sequential pipeline. Starting with DDI as the base dataset (Subsection 3.1), we develop a latent diffusion inpainting pipeline (Subsection 3.2) to remove lesions from healthy skin samples while preserving original skin tone and image context. Although healthy skin imagery may appear readily available, to our knowledge, there is no dataset with dermatologist-verified skin tone labels collected in a setting analogous to that of diseased samples. These healthy synthetics are ideal for multiple purposes: as recipient canvases for our non-parametric lesion mapping algorithm (Subsection 3.3), which places real lesion observations onto healthy skin, and as prior preservation anchors for our parametric semantic generation approach (Subsection 3.4). We generate 309 healthy, 80,427 lesion-mapped, and 185,400 semantic synthetic images. We design a protocol to verify and triage generated data via manual review and ablation, dropping between 7% and 22% of synthetics depending on the method. Full details of the review criteria and final skin-tone balance are included in the Appendix. The modular design allows each method to address different data scarcity scenarios: the lack of clinical-quality baselines, extremely rare conditions, and parametric generation that scales efficiently for cases with limited but sufficient training examples.

### 3.1   BASE DATA

We exclusively leverage DDI in our experiments as it is biopsy and dermatologist confirmed, both in terms of condition and skin tone, which avoids noise in our training process. DDI includes a range of rare, common, benign, and malignant skin lesions across all skin tones. The images were captured under variable conditions with standard RGB cameras, representative of the first-pass examination. In total, the dataset contains 656 samples, 171 malignant and 485 benign. There are 78 unique disease labels, some of which are closely related or sub-groups of each other, which we join together to a total of 65 unique disease categories, as is common in some previous work (Tschandl, 2018). We call the dataset with 65 disease categories "Joined DDI" and detail our grouping reasoning in the Appendix. After joining, 25 diseases are represented by a single observation, 27 contain 2-10 samples, and 13 diseases are captured by more than 10 observations. This becomes our base dataset.

### 3.2   LATENT DIFFUSION INPAINTING

To convert DDI into healthy samples, we perform inpainting using the frozen UNet denoiser (1.22 B parameters) and MoVQGAN decoder (67 M parameters) (Razzhigaev et al., 2023), guiding the generation via positive *"A close-up clinical photograph of healthy, smooth, normal human skin"* and negative *"Bad anatomy, deformed, lesion, ugly, disfigured, illness, hole, transparent, eye"* prompts, which avoids artifacts common when using these methods. We ablate on different prompts and prompt structures, or generation without prompts, which led to artifacts that we discuss in the Appendix. We determine the precise regions of the lesion (and the marker, if present) to inpaint using semantic segmentation masks. Finally, we apply a light Gaussian blur to the lesion and marker masks to ensure a smoother transition to healthy skin. This method is visualized in Figure 1 color-coded as green.

**Healthy Synthetics**   For masking, we elect to use sDDI as it is human-verified and delineates the skin, ruler, marker, and lesion boundaries. However, our framework is compatible with accurate out-of-the-box segmentation algorithm output masks, as shown on Section A.1. Out of a possible 334 samples (number of masks), we keep 309 healthy synthetics, discarding 25 samples (7%) after human review. This review excludes any images that exhibit unnatural artifacts, as detailed in the Appendix. We observe realistic lesion-less reconstruction of the normal expected skin under lesion or marker-covered regions, preserving the skin tone of the input sample, hair coverage, and body location. This ensures that all other original descriptors remain accurate. The results are shown in Figure 3, with additional examples in the Appendix.

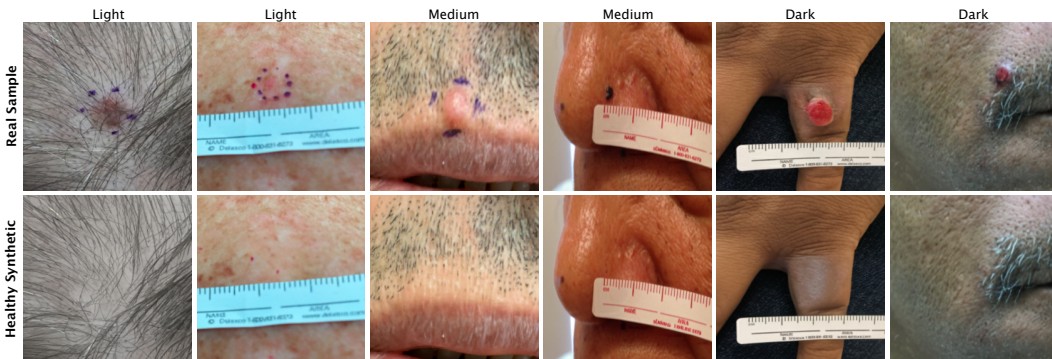

Figure 3: **Human-verified healthy synthetic imagery**. Our inpainting framework removes the target lesion (and any marks made by the marker, if present). We observe lesion-less reconstruction robust to hair, lesion morphology, or location, which ensures skin tone descriptors remain accurate. These clinical-setting healthy samples are later leveraged toward lesion-mapped and semantic synthetic images.

### 3.3 LESION MAPPING ALGORITHM

Mapping existing lesion observations from one sample to another relaxes the need to learn an accurate distribution of disease appearance and morphology, which is useful when there is not enough training data, such as in the case of rare diseases. However, non-parametric approaches risk placing lesions out of bounds, in differing lighting conditions, or with unnatural appearance.

Our lesion mapping algorithm alleviates these concerns by editing sDDI masks by joining previously "lesion" or "marker" class areas into healthy skin segments, matching our healthy synthetics. By avoiding any background or ruler sections, we explicitly determine valid locations on which a lesion can be placed. An additional padding parameter determines the minimum distance from mask edges a lesion edge must lie, which helps avoid mapping lesions onto the curvature of body parts like arms, legs, or fingers. Finally, size, rotation, and position parameters can be set or left at random. This algorithm is visualized in Figure 1, color-coded gray.

**Lesion-mapped Synthetics**    For each healthy image, we iterate on all DDI samples with sDDI masks (donors). We dilate, then apply a light Gaussian blur to the disease mask area before mapping it onto the healthy sample. We determine a random location with padding (10 pixels from the edge). If the padding algorithm does not find enough room to place the lesion mask (as can be the case for large lesions mapped to narrow body sections), we skip it and move on to the next donor. Given 309 synthetic healthy images, 334 sDDI masks, we generate a total of 80,427 lesion-mapped samples. This total is controllable as the loop could be run multiple times with different parameters. The results are shown in the second row of Figure 2, with more examples in the Appendix.

### 3.4 PARAMETRIC GENERATION

When sufficient per-disease data is available, learned approaches for photorealistic generation and fine-grain control can be trained with high performance. As our base data is small, efficient fine-tuning algorithms and pre-trained models become our starting point. We begin by learning disease-specific tokens through textual inversion (Gal et al., 2023). These special tokens are injected into prompts used to fine-tune the latent DM backbone (Rombach et al., 2022) via Low-Rank Adaptation (LoRA) for parameter-efficient learning (Hu et al., 2022). However, unlike prior work, we cover the full spectrum of skin tones, train on the smaller but more accurate Joined DDI base data, learn a greater number of unique diseases, and leverage healthy synthetics towards Prior Preservation Loss (PPL).

PPL was introduced in (Ruiz et al., 2023) to address two major issues with fine-tuning DMs: forgetting semantic knowledge (drift) and reduced output diversity. Semantic drift is the decreasing ability of the network to generate class data for which a target instance belongs to (in our case, "an image of eczema" is the class while "an image of eczema *on dark skin*" is an instance). Reduced output diversity, in our case, can limit the generation of the target from novel viewpoints, body location, skin

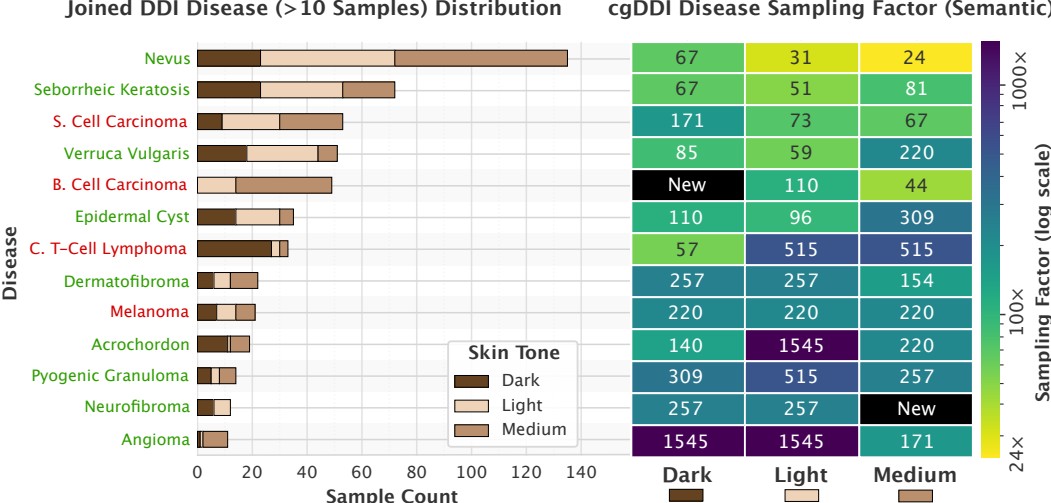

Figure 4: **(Left)** Number of observations per disease and skin-tone distribution in DDI. **(Right)** Semantic sampling factor per disease and skin-tone applied to balance the distribution up to a target number of samples (4,635 total per disease). "New" indicates that no sample of that class originally existed in DDI; cgDDI synthesized novel views up to the total balance target.

tones, etc. Recent work has found that PPL acts as a regularizer, enabling more faithful generation in comparison to textual inversion (Zeng et al., 2024) alone. We use our healthy synthetics as PPL, encouraging accurate generation with less risk of overfitting. This process is visualized in Figure 1, color-coded as purple.

**Semantic Synthetics** Since the number of samples needed to achieve quality results using our design is not clear a priori, we ablate the number of images needed to achieve viable quality generation. We do this by training on diseases in Joined DDI with two or more samples. We keep 10% of the available data per-disease (or at minimum 1 sample) for testing. At generation time, let:

$$H = \{h_i\}_{i=1}^{309}, \qquad D = \{d_j\}_{j=1}^{40}, \qquad S = \{s_k\}_{k=1}^{3} \qquad (1)$$

denote our sets of healthy images, diseases, and skin tones. For each triple $(h_i, d_j, s_k)$ we draw $R = 5$ independent semantic samples via:

$$x_{i,j,k}^{(r)} = f_{\theta,j}\Big(h_i, \underbrace{\textit{"An image of } S_{*,j} \textit{ on a } s_k\textit{-toned individual"}}_{\text{textual prompt}}; \alpha, \beta, t\Big) \qquad (2)$$

where $f_{\theta,j}$ is the disease-specific DM, $S_{*,j}$ is the special token learned by inversion, $\alpha$ the strength factor between conditioning and pure generation, $\beta$ being the guidance scale, and $t$ the number of inference steps. Optionally, we can append "with a ruler" to the textual prompt in order to encourage a measurement overlay.

The total semantic synthetic images sampled is $|H| \times |D| \times |S| \times R = 185,400$, resulting in a balanced sampling of 1,545 samples per skin tone, per disease. We empirically observe around 10 samples to be the minimum in order to train viable generators, but publish all imagery (along with the number of training samples) for further study. The results are shown on rows 3 and 4 of Figure 2. The 13 conditions with more than 10 training samples are shown on the left side of Figure 4 with a visualization of the sampling factor needed for balanced semantic synthetics on the right. Additional discussion on training efficiency and detailed parameter settings can be found in the Appendix.

## 4 CLASSIFICATION AND FAIRNESS

We evaluate the quality of our synthetic data through malignancy classification experiments evaluated on real data. For classifier design, we follow the contrastive disentanglement approach from (Du

Table 2: **Malignancy Classification and Fairness Performance.** Abbreviations: "R." denotes real data, "S." denotes synthetic data, "R. + S." denotes a combination of both.

| | Accuracy (%) $\pm$ Std-dev. | | | | Fairness $\pm$ Std-dev. | | |
|---|---|---|---|---|---|---|---|
| Method | Mean | Light | Medium | Dark | PQD | DPM | EOM |
| Baseline (R.) | $82.4 \pm 1.5$ | $83.3 \pm 1.0$ | $74.6 \pm 5.7$ | $89.7 \pm 2.2$ | $77.0 \pm 1.9$ | $\underline{75.2} \pm 13.3$ | $58.7 \pm 4.3$ |
| FairDisCo (R.) | $83.8 \pm 0.4$ | $88.6 \pm 0.1$ | $71.7 \pm 2.2$ | $92.0 \pm 2.8$ | $78.0 \pm 4.5$ | $72.8 \pm 12.0$ | $63.7 \pm 3.5$ |
| PatchAlign (R.) | $\underline{87.4} \pm 1.2$ | $\underline{89.6} \pm 2.6$ | $80.3 \pm 5.7$ | $\underline{92.3} \pm 1.3$ | $86.9 \pm 6.1$ | $74.9 \pm 12.0$ | $69.6 \pm 1.7$ |
| Exp. 1 (S. only) | $86.4 \pm 1.0$ | $88.9 \pm 1.5$ | $\underline{84.1} \pm 2.6$ | $86.0 \pm 1.8$ | $\mathbf{94.6} \pm 3.1$ | $\mathbf{82.0} \pm 9.7$ | $\underline{81.9} \pm 2.8$ |
| Exp. 2 (R. + S.) | $\mathbf{90.9} \pm 1.3$ | $\mathbf{93.3} \pm 2.2$ | $\mathbf{86.4} \pm 4.1$ | $\mathbf{93.0} \pm 1.0$ | $\underline{92.5} \pm 2.5$ | $68.8 \pm 11.3$ | $\mathbf{86.6} \pm 1.9$ |

et al., 2023) with the patch-alignment improvements in (Aayushman et al., 2024). Keeping the same classifier design and training method allows us to directly measure the benefit of our generation framework and synthesized data. We evaluate on the same DDI benchmark as (Du et al., 2023; Aayushman et al., 2024), and report the same fairness metrics, which we formalize next.

## 4.1 Classification Fairness Metrics

The fairness metrics used in our evaluation are: Predictive Quality Disparity (PQD), Demographic Disparity Metric (DPM), and Equality of Opportunity Metric (EOM). In brief, PQD measures the prediction quality difference between each skin tone group as a "best vs worst" ratio measurement, similar to IR. DPM computes the percentage diversities of positive outcomes for each skin tone group and increases as we reach similar positive prediction rates across populations. EOM measures true positive rate consistency and increases as skin tone groups have similar true positive rates. Formally:

$$\text{PQD} = \frac{\min_{k \in S} \text{acc}_k}{\max_{k \in S} \text{acc}_k} \qquad \text{DPM} = \frac{1}{|\mathcal{C}|} \sum_{c \in \mathcal{C}} \frac{\min_{k \in S} p(\hat{y} = c \mid s = k)}{\max_{k \in S} p(\hat{y} = c \mid s = k)} \qquad (3, 4)$$

$$\text{EOM} = \frac{1}{|\mathcal{C}|} \sum_{c \in \mathcal{C}} \frac{\min_{k \in S} p(\hat{y} = c \mid y = c, s = k)}{\max_{k \in S} p(\hat{y} = c \mid y = c, s = k)} \qquad (5)$$

Where $\mathcal{C} = \{\text{benign}, \text{malignant}\}$, $y$ is the ground truth and $\hat{y}$ is the model prediction.

## 4.2 Malignancy Classification

We run two experiments: *Experiment 1:* Training purely on synthetic data generated by the cgDDI framework, and *Experiment 2:* Training on synthetic data and then fine-tuning on a subset of real DDI data. Evaluations for both experiments are performed on holdout test sets of real DDI images. This is visualized in Figure 1, color coded as red. In order to avoid data leaks, we exclude training on synthetics that were conditioned on samples found in the testing set. We also avoid semantic synthetics generated with $\leq 10$ training samples.

We find that *Experiment 1* (training on synthetic cgDDI data) achieves a mean accuracy similar to the prior methods from (Du et al., 2023; Aayushman et al., 2024), while improving in fairness metrics. In *Experiment 2* (training on synthetic data and fine-tuning on real images), we achieve higher accuracy across all skin tone categories. Results of both experiments are shown in Table 2.

As stated in (Aayushman et al., 2024), EOM is the most important fairness metric, and *Experiment 2* achieves the highest EOM score among all the baseline methods, followed by *Experiment 1*. This improvement in the EOM metric demonstrates the essential role of using synthetically generated data from cgDDI in the training regimen. We also observed that the PQD metric is slightly lower in *Experiment 2* than in *Experiment 1*, as the real-data performance gains appear to be greater for light and dark skin tones than for medium. This could be due to disease imbalances found in the real data. While DPM decreases in *Experiment 2*, we note that metric can increase with false positives, as noted by (Aayushman et al., 2024).

Table 3: **Left:** classification accuracy by rarity. **Right:** generative quality by skin tone.

| Disease Commonality | Exp. 1 (S) Accuracy (%) | Exp. 2 (S+R) Accuracy (%) | Skin Tone | FID $\downarrow$ | KID $\downarrow$ | LPIPS $\downarrow$ |
|---|---|---|---|---|---|---|
| | | | Light | 103.45 | 0.039 | 0.715 |
| Common (>10) | 85.05 | 91.59 | Medium | 88.41 | 0.032 | 0.741 |
| Rare (3–10) | 94.74 | 89.47 | Dark | 108.07 | 0.016 | 0.734 |
| V. Rare (1–2) | 83.33 | 83.33 | Max/Min | 1.22 | 2.41 | 1.04 |

### 4.2.1 DISEASE RARITY PERFORMANCE

To understand how synthetic augmentation impacts diseases with varying training data availability, we stratify classification performance by disease frequency. We define common diseases as those represented by more than 10 observations (107 test cases), rare as between 3 and 10 (19 test cases), and very rare as between 1 and 2 (6 test cases).

We observe that fine-tuning on real data improves the common case, slightly reduces the rare case, and has no impact on the very rare case. As discussed, our synthetic dataset contains significantly more observations of rare diseases compared to base DDI, while the intricacies and imbalance of real data encourage the model to improve the common case (leading to a higher mean accuracy) as shown on Table 3 (Left). Notably, our lesion-mapping approach enables augmentation for single-sample diseases, maintaining competitive accuracy even for conditions with minimal training data.

### 4.3 GENERATIVE FAIRNESS METRICS

While classification fairness metrics demonstrate improved equity in malignancy diagnosis, we additionally evaluate whether our generation methods maintain quality parity across skin tones. We compute Fréchet Inception Distance (FID) (Heusel et al., 2017), Kernel Inception Distance (KID) (Binkowski et al., 2018), and Learned Perceptual Image Patch Similarity (LPIPS) (Zhang et al., 2018) between cgDDI and held-out real DDI images.

All three metrics demonstrate reasonable stability across skin tones, with low dispersion for FID ($\sigma = 8.39$, max/min ratio = 1.22) and LPIPS ($\sigma = 0.011$, max/min ratio = 1.04). KID shows the largest relative spread ($\sigma = 0.010$, max/min = 2.41), but the absolute differences are small, so the practical impact is limited. Each metric slightly favors a different tone (FID: Medium best, KID: Dark best, LPIPS: Light best), indicating no systematic advantage for any population. This is shown in Table 3 (Right). We further ablate these metrics per generative method in the Appendix, along with additional visual examples that reinforce generative fairness across populations.

## 5 CONCLUSION

We present cgDDI, a novel hybrid generation framework that solves the technical challenge of synthesizing fair and diverse dermatological imagery under extreme data constraints. A key contribution in combining non-parametric lesion mapping (enabling $300\times$ augmentation of single-sample diseases) with parametric generation (achieving high-fidelity synthesis from just 10 samples). Additionally, the method is capable of generating in-distribution healthy images which are later leveraged through the pipeline. This approach maintains quality parity across skin tones while providing fine-grained semantic control for both common and rare diseases.

The method effectiveness is validated through large data synthesis and extensive experiments: models trained purely on our synthetic data achieve competitive accuracy, and state-of-the-art when fine-tuned on real data. Additionally we measure generative fairness with low disparity between skin-tones and strong performance for very rare diseases. By successfully growing a small but clean medical base dataset, we demonstrate careful algorithmic design can overcome severe data bias and limitations in medical AI and potentially other data-constrained domains. We further discuss generalizability in the Appendix, noting that while we leverage DDI due to its unique properties, all components of our design are applicable to other datasets.

ETHICS STATEMENT

**Human subjects, privacy, and de-identification:** Our experiments utilize de-identified (no faces, tattoos, or other Personally Identifiable Information (PII)), biopsy-confirmed macroscopic clinical photographs with dermatologist-provided skin-tone labels within DDI, collected under Stanford IRB protocols 36050 and 61146. We do not collect new human-subjects data. Manual review and discard criteria for synthetics was performed by two researchers in the Medical AI field (paper authors) with the following demographics: one hispanic latino man and one middle-eastern woman.

**Fairness and potential bias:** The central motivation is improving equity in dermatological AI: cgDDI explicitly balances across Fitzpatrick I–VI and augments extremely rare conditions. We quantify generative quality across tones and evaluate classifier fairness with tone-stratified metrics. Residual risks include distributional shift and synthetic artifacts that could differentially impact subgroups; we mitigate via human triage, mask-quality protocols, and stratified analyses.

**Artifacts:** We do not distribute clinical images, we release only our original artifacts: synthesized data, models and code under CC BY-NC. Fine-tuned checkpoints are allowed to be re-parametrized. Our usage terms restrict unethical use. We provide contact/takedown mechanisms. Our work is transformative non-commercial research use; we welcome coordination with data stewards to ensure ongoing compliance. Shared croissant metadata includes all relevant Responsible AI fields.

**Dual-use and clinical use:** Synthetic medical images could be misused for deceptive content or overconfident automated diagnosis. We provide safeguards: clear labeling of synthetic data and strict intended-use terms. No clinical deployment is claimed.

**Conflicts of interest and sponsorship:** No conflicts or external sponsors influenced this research.

Our work aims to democratize access to fair dermatological AI research, particularly benefiting under-resourced communities where algorithmic fairness is most critical. We emphasize that synthetic data should augment, not replace, real datasets and classification systems should augment, not replace, clinician validation.

REPRODUCIBILITY STATEMENT

We ensure full reproducibility through comprehensive resource release and documentation.

**Code and Models:** Complete implementation available on `https://anonymous.4open.science/r/ControllableGenDDI` including: all generation pipelines with hyperparameters, training scripts for textual inversion and LoRA fine-tuning, classification architecture from prior work (Aayushman et al., 2024), 65 disease-specific pre-trained model weights. Note that models are to be released upon publication given difficulties in anonymization, however we provide the necessary information to recover these weights.

**Data:** We share 266,136 synthetic images on a anonymous bucket linked in `https://anonymous.4open.science/r/ControllableGenDDI` with complete metadata (skin tone, disease, generation method, prompts) and croissant file. A more formal and organized repository will be shared upon acceptance.

**Experimental Setup:** Five-fold cross-validation using random seeds from (Aayushman et al., 2024). Hardware specifications: NVIDIA L4 24GB GPUs, Intel Xeon CPUs, 53GB RAM, Software environment: Ubuntu 22.04.4, CUDA 12.4, Python 3.11.12. Full details available in the Appendix.

**Evaluation:** Fairness metrics computation code provided. Segmentation masks from sDDI (Carrión & Norouzi, 2023) required to replicate healthy generation. All experiments are reproducible using provided code and specified dependencies.

**Specifications:** All relevant sampling parameters, hyper-parameters, algorithmic settings and details needed to replicate our results are within the manuscript, appendix or attached code repository.

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

Figure 5: **Fitzpatrick17k Algorithmic Masking and Healthy Sampling**. We expand our work to other datasets which may not include lesion and skin masks. Masks are generated using an off-the-shelf segmentation model.

# APPENDIX

# A    ADDITIONAL DATASET VALIDATION

We discuss additional real datasets relevant to our study in Section 2, particularly, Fitzpatrick17k (F17k) as the macroscopic dataset most commonly cited when training and evaluating fairness in dermatological AI. However, we note F17k is skin-tone unbalanced, annotated by non-experts without biopsy or histopathological evidence, leading to more than $30\%$ label noise Groh et al. (2021). To evaluate the generalizability of our method and to demonstrate mask-free processing, while reducing these concerns, we run the full cgDDI framework on a verified subset of F17k Groh et al. (2024).

This dataset totals 364 samples: 143 light-skinned, 113 medium-skinned and 108 dark-skinned. The samples cover eight main diseases: atopic dermatitis, cutaneous T-cell lymphoma, dermatomyositis, lichen planus, lyme disease, pityriasis rosea, pityriasis rubra pilaris and secondary syphilis. Only cutaneous T-cell lymphoma is malignant and overlaps with DDI. We begin processing this data by generating masks with an off-the-shelf segmentation algorithm.

## A.1    AUTOMATED SEGMENTATION MASKS

When processing the DDI dataset, we elect to leverage pre-made expert-verified segmentation masks Carrión & Norouzi (2023) as they are readily available for this dataset. However, when processing other datasets, this sort of additional information may not be included. We mention in Section 3.2 that our framework is compatible with algorithmically generated masks, and, in this section we demonstrate this functionality by leveraging the Segment Anything 3 (SAMv3) model Carion et al. (2025). We elect this model as a generalist segmentation algorithm proof-of-concept, we acknowledge other methods, particularly those tailored toward dermatology, might perform equally or better.

## A.2    HEALTHY SYNTHETICS: F17K

We perform a segmentation inference pass tasking SAMv3 with predicting "rash" areas with a confidence cutoff of $0.35$. Subsequently we input these masks into Section 3.2 of our framework as-is, outputting healthy F17k synthetics which will serve toward the rest of our pipeline. As with DDI, we manually verify all generated healthy samples and keep those passing the discard criteria outlined in Section H.1. After review, 46 verified healthy samples are kept: 21 light-skinned, 16 medium-skinned and 9 dark-skinned. This discard rate is higher than for DDI, we believe this is due

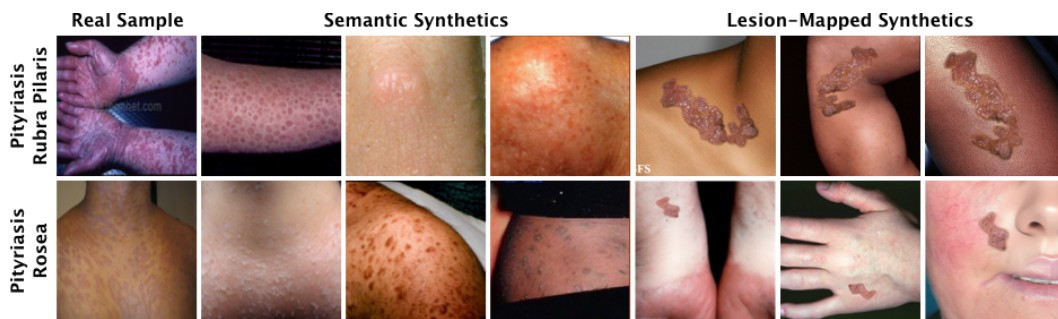

Figure 6: **Fitzpatrick17k Semantic and Lesion-Mapped Synthetics**. Semantic sampling captures the intricacies differentiating these two related diseases as we LoRA fine-tune separate models per-disease while mapping extends the views for each real lesion.

to imperfect masks leading to a larger number artifacts. Results for this and Section A.1 are shown on Figure 5.

### A.3   LESION-MAPPED SYNTHETICS: F17K

We perform a second segmentation inference pass tasking SAMv3 with delineating the "skin" area of our synthetic healthy samples with a confidence cutoff of 0.35. The aforementioned lesion masks and these new skin masks are passed into Section 3.3 of our pipeline as-is. The discard criteria described in Section H.2 is applied (46.9% discard rate), leading to 1,124 new lesion-mapped synthetics: 529 light-skinned, 373 medium-skinned, 222 dark-skinned samples. The larger imbalance observed here is largely due to F17k skin-tone distribution and the resulting healthy synthetic output distribution.

### A.4   SEMANTIC SYNTHETICS: F17K

We now randomly sample 85% of the verified F17k dataset to learn text-inversion tokens for each of the conditions, then fine-tune generative models using LoRA and prior preservation via healthy synthetics following Section 3.4. The remaining 15% of the data is hold-out for classifier testing, and is balanced across skin-tones. Relevant to the minimum training data and discard criteria outlined on Section H.3, all 8 main conditions in F17k contain more than 10 training samples. We then sample 690 (46 healthy images x 3 skin tones x 5 samples) semantic synthetics per condition, for a 5,520 total new images equally split across skin-tones (1,840 each). Results for this and Section A.3 are shown on Figure 6.

### A.5   CLASSIFICATION FAIRNESS: F17K

We repeat classification experiments using the PatchAlign Aayushman et al. (2024) method described in Section 4 and compute the fairness metrics shown on Section 4.1. These experiments are tested on the held-out F17k data described above. We first train a baseline model, utilizing the same classifier algorithm and training parameters on real data (specifically, the 85% used above for training the generative models). This baseline method reaches 86.0% classification accuracy. We then train from the same initialization but purely on F17k lesion-mapped and semantic synthetics, achieving 88.4% mean accuracy (up 2.4%), with a rise in medium skin-tone accuracy which is offset by lower dark-skin accuracy with fairness metrics being relatively close. We then take this checkpoint and fine-tune on real data, we observe mean accuracy rises another 2.3% to 90.7% and leading or tied per skin-tone accuracy for all skin-tones. PQD also is also highest here and while DPM is lower than the baseline method, Aayushman et al. (2024) argues that PQD is the most important metric while DPM can be skewed by false positives as discussed on the main text. These results are shown on Table 4.

## B   CROSS-DATASET VALIDATION

In this section, we emphasize cross-data framework compatibility by mapping and sampling synthetics using combination inputs from both F17k and DDI datasets.

Table 4: F17k-trained models evaluated on the F17k test set. Baseline is trained on real data.

| Setting | Method | Acc | Light | Med | Dark | PQD | DPM | EOM |
|---------|--------|-----|-------|-----|------|-----|-----|-----|
| F17k → F17k | Baseline | 86.0% | 86.7% | 82.4% | **90.9%** | 0.906 | **0.455** | 0.500 |
| | Synth Only | 88.4% | 86.7% | **94.1%** | 81.8% | 0.869 | 0.441 | 0.500 |
| | Synth + Real | **90.7%** | 86.7% | **94.1%** | 90.9% | **0.921** | 0.441 | 0.500 |

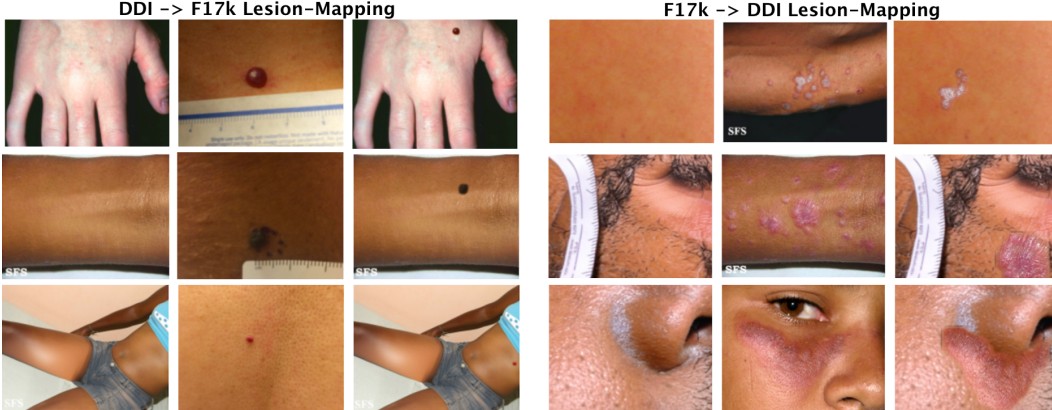

Figure 7: **Bidirectional Lesion-Mapping**. We leverage our method as-is to map lesions from one dataset to the other and vice versa.

## B.1 CROSS-DATASET LESION MAPPING

We demonstrate lesion-mapping generalizability through bidirectional cross-dataset synthesis, in this case, we show effective mapping of lesions from DDI to F17k and vice versa. This does not require additional processing or changes other than swapping the directory of source and target samples. The resulting synthetics are visualized on Figure 7. The resulting totals after discard criteria outlined on Section H.2 is shown on Table 5. Given that DDI is more balanced per skin-tone we see a closer distribution when mapping to it compared to F17k.

Table 5: Cross-dataset Lesion-Mapping

| Direction | Total | Light | Medium | Dark |
|-----------|-------|-------|--------|------|
| DDI lesions → F17k healthy | 13,822 | 6,593 | 4,267 | 2,962 |
| F17k lesions → DDI healthy | 9,394 | 3,103 | 3,272 | 3,019 |

## B.2 CROSS-DATASET SEMANTIC GENERATION

Similarly, we leverage the previously trained generative models and sample new synthetics using the cross-dataset healthy images as image prompts. Note that these models are not re-trained (as there is little disease overlap between datasets) but simply re-prompted with new image samples. Table 6 displays the total number of new semantic synthetics. As skin-tone sampling here is controlled via textual prompts we are able to equally sample across target skin-tones.

## B.3 CROSS-DATASET CLASSIFICATION EXPERIMENTS

In order to exhaustively test the cross-dataset classification capabilities of individually and in-combination trained models, we run extensive experiments detailing the overall accuracy, per skin-tone accuracy and fairness metrics for various configurations, keeping the base classifier we have used through the paper based on Aayushman et al. (2024).

Table 6: Cross-dataset Semantic Sampling

| Configuration | Total Images | Per Skin Tone |
|---|---|---|
| F17k diseases + DDI healthy prompts | 37,080 | 12,360 |
| DDI diseases + F17k healthy prompts | 8,970 | 2,990 |

Table 7 shows the performance of F17k-trained models on fully unseen DDI data and DDI-based model performance on fully unseen F17k data. Note that when testing a DDI trained model on F17k or vice versa the little overlap between diseases might mean the model has not generalized well to unseen disease classes. This leads to the first F17k baseline classifier obtaining higher accuracy but lower fairness metrics. When training on DDI synthetics however, the richer, larger, balanced and more diverse set of synthetics generalize significantly better to unseen F17k data.

Table 7: Cross-dataset transfer performance, a model is trained on one dataset then evaluated on the other's test set. Baseline model is trained solely on real data.

| Setting | Method | Acc | Light | Med | Dark | PQD | DPM | EOM |
|---|---|---|---|---|---|---|---|---|
| F17k → DDI | Baseline | **79.6%** | **64.3%** | **82.2%** | **87.5%** | 0.735 | 0.500 | 0.500 |
| | Synth Only | 74.3% | 57.1% | 77.8% | 82.5% | 0.693 | 0.604 | 0.440 |
| | Synth + Real | 75.2% | **64.3%** | 80.0% | 77.5% | **0.804** | **0.678** | **0.703** |
| DDI → F17k | Baseline | 60.5% | 66.7% | 47.1% | 72.7% | 0.647 | 0.515 | 0.526 |
| | Synth Only | **74.4%** | **80.0%** | **70.6%** | 72.7% | **0.882** | 0.556 | **0.613** |
| | Synth + Real | 69.8% | 73.3% | 76.5% | 54.5% | 0.713 | **0.664** | 0.388 |

## B.4    AGGREGATED CROSS-DATASET TRAINING

We also train on a mixed collection of both datasets, baseline using only real data, using purely synthetics and also fine-tuning those checkpoints on real imagery. Table 8 presents the complete aggregated results. We train on the full combination of DDI and F17k synthetics, including within-dataset lesion-mapped and semantic samples as well as cross-dataset variants from Tables 5 and 6, then fine-tune on the aggregated real training data from both sources. We observe the synthetics and real data generally outperform other settings, highlighting the value of data augmentation using our methods.

Table 8: Aggregated cross-dataset performance. Baseline model is trained on mixed real data.

| Setting | Method | Acc | Light | Med | Dark | PQD | DPM | EOM |
|---|---|---|---|---|---|---|---|---|
| Mix → F17k | Baseline | 86.0% | 86.7% | 88.2% | 81.8% | **0.927** | **0.471** | 0.500 |
| | Synth Only | 86.1% | **93.3%** | 88.2% | 72.7% | 0.779 | 0.378 | **0.750** |
| | Synth + Real | **93.0%** | **93.3%** | 88.2% | **100.0%** | 0.882 | 0.378 | 0.500 |
| Mix → DDI | Baseline | 83.2% | 75.0% | 82.2% | 90.0% | 0.833 | **0.679** | 0.784 |
| | Synth Only | 79.7% | 71.4% | 84.4% | 80.0% | 0.846 | 0.436 | **0.833** |
| | Synth + Real | **86.7%** | 82.1% | 84.4% | 92.5% | **0.888** | 0.600 | 0.750 |

## C    F17K ADDITIONAL EXPERIMENTS DISCUSSION

The cross-dataset experiments demonstrate several key properties of the cgDDI framework.

**Pre-made segmentation masks are not necessary.** We observe out-of-the-box segmentation algorithms are capable of fair and accurate masks which are then directly compatible with the cgDDI framework. Using pixel-perfect and verified masks might lead to a lower discard rate but this trade-off is likely worth it for the scalability of algorithmically generated masking.

**Effective cross-dataset transfer.** Lesion-mapping algorithms and generative models can be effectively leveraged cross datasets, leading to higher diversity synthetics which lead to improved classifier performance downstream. For example, training exclusively on real F17k data we achieve 86.0% accuracy, adding synthetics from our method raises this metric to 90.7% while finally mixing in DDI synthetics achieves 93.0%.

**Improved fairness through dataset aggregation.** We observe the leading PQD score includes our synthetics in five out of six experiments. The only exception being a large mix of real training data which highlights the persistent value of clean, high-quality real samples.

**Knowledge transfer despite minimal disease overlap.** Only one condition (cutaneous T-cell lymphoma) appears in both DDI and the verified F17k subset. Nevertheless, cross-dataset training improves performance on both benchmarks, suggesting the model learns generalizable features about skin lesion appearance rather than dataset-specific patterns.

# D    APPLICABILITY TO ADDITIONAL DATASETS

Our framework's design principles are dataset-agnostic and applicable to any medical imaging domain with segmentation masks:

## D.1    COMPONENT TRANSFERABILITY

- **Inpainting Pipeline:** Compatible with any medical images where lesion removal is meaningful (dermatology, ophthalmology, radiology)
- **Lesion Mapping:** Generalizes to any scenario requiring transplantation of pathological regions between images
- **Parametric Generation:** Standard diffusion fine-tuning applicable to any labeled collection with $\geq 10$ samples

## D.2    DOMAIN-SPECIFIC CONSIDERATIONS

While we validate on dermatology due to its unique fairness challenges and data availability, adaptation to other domains requires:

1. Segmentation masks (manual or automated via SAM/DINO)
2. Fairness-relevant metadata (demographics, equipment type, acquisition parameters)
3. Domain expertise for quality assessment

## D.3    DDI SELECTION

We selected DDI over other datasets due to their (1) lack of biopsy confirmation leading to unreliable ground truth (F17k), (2) absence of dermatologist-verified skin-tone labels (SCIN) needed for fairness evaluation, (3) the dermoscopic setting being relatively unrepresentative of clinical practice in very low resource communities (HAM1000) (4) the ready availability of segmentation masks for DDI (sDDI) and (5) computational constraints preventing exhaustive cross-dataset experiments within the review period. Due to these limitations growing the complexity of further experiments on other datasets, we leave these experiments as future work.

# E    DDI PREPARATION AND GROUPING

This section details our dataset preparation process, including the grouping methodology for disease categories and the mask annotation protocol.

## E.1    JOINED-DDI CATEGORY GROUPING

In our main manuscript, we mention consolidating 78 unique disease labels in the original DDI dataset into 65 unique disease categories in what we call "Joined DDI." This grouping was performed based on histopathological similarity, diagnostic similarities, and standard dermatological practice.

For example, several variants of Basal Cell Carcinoma (BCC) were grouped into a single BCC category, including basal-cell-carcinoma, basal-cell-carcinoma-superficial, and basal-cell-carcinoma-nodular. Similarly, different types of melanoma (melanoma-in-situ, melanoma-acral-lentiginous, nodular-melanoma) were consolidated into a single melanoma category, while maintaining distinctions between fundamentally different malignancies. Table 9 shows representative examples of our grouping strategy.

Table 9: Representative examples of disease grouping from original DDI labels to Joined DDI categories

| Original DDI Labels | Joined DDI Category |
| --- | --- |
| basal-cell-carcinoma basal-cell-carcinoma-superficial basal-cell-carcinoma-nodular | basal cell carcinoma |
| melanoma-in-situ melanoma-acral-lentiginous nodular-melanoma-(nm) melanoma | melanoma |
| squamous-cell-carcinoma-in-situ squamous-cell-carcinoma squamous-cell-carcinoma-keratoacanthoma | squamous cell carcinoma |
| mycosis-fungoides subcutaneous-t-cell-lymphoma | cutaneous T-cell lymphoma |
| seborrheic-keratosis-irritated seborrheic-keratosis | seborrheic keratosis |
| melanocytic-nevi dysplastic-nevus | nevus |
| verruca-vulgaris wart | verruca vulgaris |
| benign-keratosis inverted-follicular-keratosis | benign keratosis |
| atypical-spindle-cell-nevus-of-reed pigmented-spindle-cell-nevus-of-reed | spindle cell nevus of Reed |

Our disease grouping strategy reduced the number of unique disease categories while preserving clinically relevant distinctions. This approach provided more samples per category for training our generative models while maintaining the diagnostic utility of the dataset. The complete mapping includes 78 original DDI disease labels consolidated into 65 Joined DDI categories, with the remaining categories having a one-to-one mapping between original and joined labels.

### E.2 MASK ANNOTATIONS

Our work builds upon the segmentation masks from sDDI (Carrión & Norouzi, 2023), which provide pixel-level annotations for four classes: skin, lesion, ruler, and marker. We applied the following post-processing steps to these masks to prepare them for our generation pipeline:

- **Boundary refinement:** To ensure smooth transitions in the inpainting process, we applied a dilation operation followed by Gaussian blur to the target mask boundaries.

- **Mask consolidation for healthy synthetics:** We merged lesion and marker classes into a single "skin" as the original lesion and marker were removed and replaced with normal skin during our inpaiting process.

- **Valid mask area:** The lesion mapping algorithm, calculates "valid region" masks by identifying skin-only regions where lesions could be realistically placed, with or without padding. This avoids placing lesions on top of the ruler area, for example.

In order to have the sDDI (Carrión & Norouzi, 2023) mask fit the image data, we also need to use the image pre-processing technique from that work. It mainly involves cropping DDI images into square shape, more detail can be found on (Carrión & Norouzi, 2023).

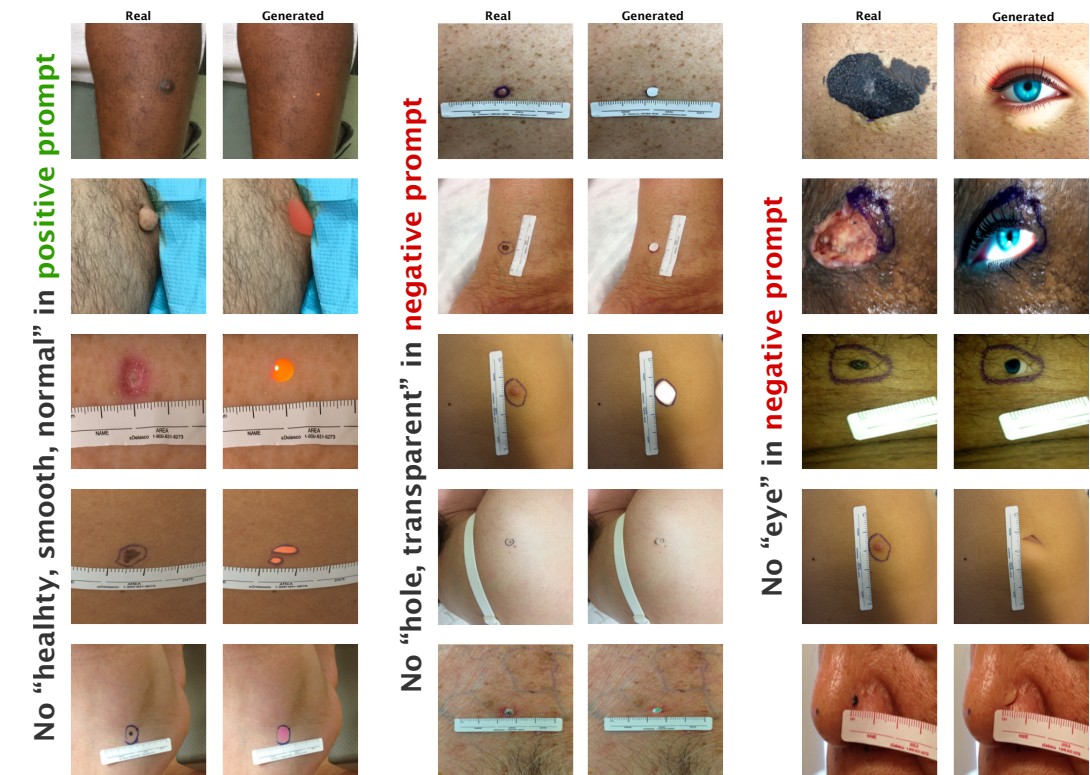

Figure 8: **Ablation of inpainting prompts**. We observe anatomically incorrect generation of healthy samples if we do not include terms like "healthy, smooth normal" as guidance on the positive prompt (left). Similarly, excluding terms from the negative prompt like "hole, transparent" (center) or "eye" (right) yield artifacts.

# F  GENERATIVE FAIRNESS ABLATIONS

## F.1  CONTRIBUTION OF GENERATION METHODS

We evaluate the individual contribution of each generation method to classification performance by training models on different subsets of cgDDI.

Table 10: Ablation study: Classification performance using different synthetic data subsets.

| Training Data | Mean Acc. | Light | Medium | Dark |
|---|---|---|---|---|
| Real DDI only | 82.4 | 83.3 | 74.6 | 89.7 |
| Healthy + Lesion-mapped | 81.2 | 82.1 | 78.4 | 82.9 |
| Semantic only | 84.7 | 86.3 | 82.5 | 85.2 |
| All cgDDI (Exp. 1) | 86.4 | 88.9 | 84.1 | 86.0 |

Training solely on healthy and lesion-mapped synthetics slightly reduces overall performance compared to training solely on DDI (-1.2% mean accuracy) however it does increase the medium skin-tone performance. We believe this is likely due the limited appearance distribution of non-lesion image features (body location, lighting conditions, skin-tone, etc) and the still relative imbalance bias originating from the base dataset toward medium skin-tone in the lesion-mapped synthetics. Semantic synthetics, however, provide strong individual contribution (+2.3% mean accuracy) and a positive boost across all skin-tones, likely due to parametric learning of disease-specific features. The combination of all three methods yields the best overall performance, growing the training distribution and validating our multi-pronged approach. These results are shown on Table 10.

## F.2 GENERATIVE QUALITY BY METHOD

We provide detailed quality metrics for each generation method in the same way as in the main text via Fréchet Inception Distance (FID), Kernel Inception Distance (KID) and Learned Perceptual Image Patch Similarity (LPIPS). We now discuss performance per generative method:

**Lesion-mapped (non-parametric):** This method achieves the strongest distributional alignment, with the best mean FID (79.73) and best mean KID (0.003). Its LPIPS (0.698) matches Healthy data. Variability across tones is moderate (FID $\sigma = 12.28$; KID $\sigma = 0.002$), consistent with the methods design for rare diseases.

**Healthy:** These metrics come in second in FID (mean 102.48) and KID (mean 0.010), while LPIPS is strong (0.698), matching Lesion-mapped in perceptual similarity. Tone-wise variability is modest for KID ($\sigma = 0.001$) and larger for FID ($\sigma = 13.35$), suggesting broadly similar behavior across tones with some room to improve distributional alignment.

**Semantic (parametric):** This approach shows the most tone consistency on FID ($\sigma = 4.10$) but has the weakest absolute FID (mean 130.52) and KID (mean 0.054). LPIPS is higher (0.746) than the other methods, indicating lower perceptual similarity. Semantic generation shows consistency across tones, despite being fully parametric while having the largest divergence from the base dataset as a learned and sampled approach.

Detailed results are shown on Table 11. These metrics complement our classification fairness results in Tables 2, 3 and the visual qualitative results throughout the paper. Additionally, this supports cgDDI maintains equitable generation quality across populations.

Table 11: FID, KID, and LPIPS scores stratified by generation method and skin tone.

| Metric | Method | Light | Medium | Dark | $\sigma$ |
|---|---|---|---|---|---|
| FID ↓ | Healthy | 94.02 | 92.11 | 121.33 | 13.35 |
| | Lesion-mapped | 72.75 | 69.46 | 96.99 | 12.28 |
| | Semantic | 134.67 | 124.94 | 131.95 | 4.10 |
| KID ↓ | Healthy | 0.012 | 0.008 | 0.010 | 0.001 |
| | Lesion-mapped | 0.005 | 0.003 | 0.001 | 0.002 |
| | Semantic | 0.055 | 0.069 | 0.037 | 0.013 |
| LPIPS ↓ | Healthy | 0.703 | 0.692 | 0.700 | 0.005 |
| | Lesion-mapped | 0.703 | 0.692 | 0.698 | 0.005 |
| | Semantic | 0.738 | 0.756 | 0.743 | 0.008 |

## G SYNTHETIC DATA GENERATION DETAILS

This section provides detailed information about our three synthetic image generation approaches: latent diffusion inpainting, non-parametric lesion mapping, and parametric semantic generation.

Table 12: **Dataset Skin-Tone Imbalance.** We measure imbalance ratio for different datasets, defined on Equation 3, where a perfectly balanced dataset has a ratio of 1.

| | Existing Real Datasets | | | cgDDI | | |
|---|---|---|---|---|---|---|
| Skin-Tone | F17k | SCIN | DDI | Healthy | Lesion-map | Semantic |
| Light (I–II) | 7,755 | 730 | 208 | 97 | 25,726 | 61,800 |
| Medium (III–IV) | 6,089 | 1,088 | 241 | 114 | 30,392 | 61,800 |
| Dark (V–VI) | 2,168 | 357 | 207 | 98 | 24,309 | 61,800 |
| Imbalance Ratio | 3.58 | 3.05 | 1.16 | 1.18 | 1.25 | **1.00** |

### G.1 SKIN-TONE BALANCE

To ensure cgDDI does not over- or under-represent any particular skin tone, we measure the Imbalance Ratio (IR), defined as:

$$\text{IR} = \frac{\max_{k \in S} n_k}{\min_{k \in S} n_k}, \tag{3}$$

where $n_k$ is the number of samples of skin-tone $k$. A perfectly balanced set has an IR of 1, with larger values indicating an increasing skew. We find a higher imbalance in the larger crowd-sourced datasets F17k (Groh et al., 2021) and SCIN (Jeong et al., 2024), with DDI (Daneshjou et al., 2022b) being the most balanced real dataset. We confirm our sampling methods have not caused a significantly increased imbalance compared to DDI and improves upon other real datasets as shown on Table 12.

### G.2 DESCRIPTORS

We record and share metadata for all samples. Each row contains the output image, text caption description, synthetic type (healthy, lesion-mapped, semantic), prompt image ID, ruler (if used), textual prompt disease, malignancy, and skin tone (if used), whether the generator was trained, and finally, the number of training samples.

### G.3 LATENT DIFFUSION INPAINTING: HEALTHY SYNTHETICS

Our inpainting leverages the UNet denoiser (1.22B parameters) and MoVQGAN decoder (67M parameters) from (Razzhigaev et al., 2023). The generation process can be formalized as:

$$x_{\text{healthy}} = h_\theta(x_{\text{original}}, m, p_{\text{pos}}, p_{\text{neg}}) \tag{7}$$

where $x_{\text{original}}$ is the original DDI image, $m$ is the binary mask indicating areas to be inpainted (lesion or marker from sDDI (Carrión & Norouzi, 2023)), $p_{\text{pos}}$ is the positive prompt guiding healthy skin generation, $p_{\text{neg}}$ is the negative prompt preventing artifacts, $h_\theta$ is the inpainting model with parameters $\theta$, $x_{\text{healthy}}$ is the resulting healthy synthetic image.

We experimented with various prompt templates to optimize inpainting quality. Figure 8 shows the different prompt combinations tested and their effects on the generated images.

Our optimal positive prompt was "A close-up clinical photograph of healthy, smooth, normal human skin" and negative prompt was "Bad anatomy, deformed, lesion, ugly, disfigured, illness, hole, transparent, eye." This combination provided the best balance of realistic skin texture while avoiding common generation artifacts.

For mask processing, we applied a dilation with kernel size 5, followed by Gaussian blur with $\sigma = 2.0$ to ensure smooth transitions between inpainted and original regions.

### G.4 NON-PARAMETRIC LESION-MAPPING ALGORITHM

Our lesion mapping approach transfers lesions from source images to healthy synthetic images while preserving realism and anatomical context. The process can be mathematically described as:

$$x_{\text{mapped}} = g(x_{\text{healthy}}, x_{\text{donor}}, m_{\text{lesion}}, m_{\text{valid}}, l, p, s, r) \tag{8}$$

where $x_{\text{healthy}}$ is the healthy synthetic image, $x_{\text{donor}}$ is the source image containing the lesion, $m_{\text{lesion}}$ is the lesion mask from the donor image, $m_{\text{valid}}$ is the valid region mask for the healthy image, $l$ is coordinate location of the lesion placement, $p$ is the padding parameter, $s$ is a lesion scaling multiplier, $r$ is a rotation parameter, $g$ is the mapping function, $x_{\text{mapped}}$ is the resulting lesion-mapped synthetic image.

We set the following parameters:

- $l$ – we allow the coordinate location of the mapped lesion to be determined at random given the padding $p$ constraint.

- $p$ – we set the padding parameter to 10 pixels. This means the edge of the lesion mask must be place at a minimum of pixels from the edge of the valid skin mask.

- $s$ – we set the scaling multiplier to 1.0, meaning no up or down-scaling was applied. We will investigate in the future how to best represent the scale range of particular diseases in order to best leverage this parameter.

- $r$ – we set the rotation parameter to zero degrees, meaning no rotation was applied. We noticed large rotations impact the natural appearance of the synthetic with relation to lighting, future work intends to automatically optimize the rotation in relation to real-world illumination.

## G.5 PARAMETRIC SEMANTIC GENERATION VIA TEXTUAL INVERSION & LORA

Our parametric generation approach learns disease-specific concepts through textual inversion and fine-tunes the latent diffusion model via Low-Rank Adaptation (LoRA). The process is formalized as:

$$S_{*,j} = \text{TextualInversion}(D_j, \phi) \tag{9}$$

$$\theta_j = \text{LoRAFineTuning}(\theta, D_j, S_{*,j}, x_{\text{healthy}}) \tag{10}$$

$$x_{i,j,k}^{(r)} = f_{\theta_j}(h_i, \text{"An image of } S_{*,j} \text{ on a } s_k\text{-toned individual"}; \alpha, \beta, t) \tag{2}$$

where $D_j$ is the set of training images for disease $j$, $\phi$ are the parameters of the text encoder, $S_{*,j}$ is the learned special token for disease. For fine-tuning $j$, $\theta$ are the base model parameters, $\theta_j$ are the LoRA-adapted parameters for disease $j$. Sampling is finally done with $h_i$ as the $i$-th healthy synthetic image, $s_k$ is the $k$-th skin tone, $\alpha$ is the strength factor, $\beta$ is the guidance scale, $t$ is the number of inference steps, $x_{i,j,k}^{(r)}$ is the $r$-th semantic synthetic sample for healthy image $i$, disease $j$, and skin tone $k$.

We set the following parameters:

- $\alpha$ – we set the strength factor to 0.725, we observe setting $\alpha$ much lower than 7.0 to struggle at guiding generation towards the prompted skin tone.

- $\beta$ – we set the guidance scale to 8.75.

- $t$ we set the number of inference steps to 100.

## G.6 TRAINING HYPERPARAMETERS

We discuss our training hyperparmters here, which can also be found on the code repository. As the inpainting pipeline is frozen it does not necessitate training hyperparameters. The lesion mapping algorithm is non-parametric, so it does not require hyperparameters.

### G.6.1 TEXTUAL INVERSION

For textual inversion, we set batch size to 4, maximum training steps to 500, learning rate to $5.0e^{-4}$, initializer token set to "skin" and training is done at "fp16" precision. We do not leverage learning rate warm-up and we set a constant learning rate schedule.

### G.6.2 LORA FINE-TUNING

For LoRA fine-tuning, we set batch size to 16, sampling batch size is also set to 16, we set one step of gradient accumulation, maximum training steps to 750, learning rate to $5.0e^{-6}$, we point to our healthy images as class images and training is done at "fp16" precision. We do not leverage learning rate warm-up and we set a constant learning rate schedule.

## G.7 ABLATIONS ON DATA EFFICIENCY

To determine the minimum number of real samples needed for effective generation, we conducted ablation studies on the relationship between training sample size and generation quality via visual

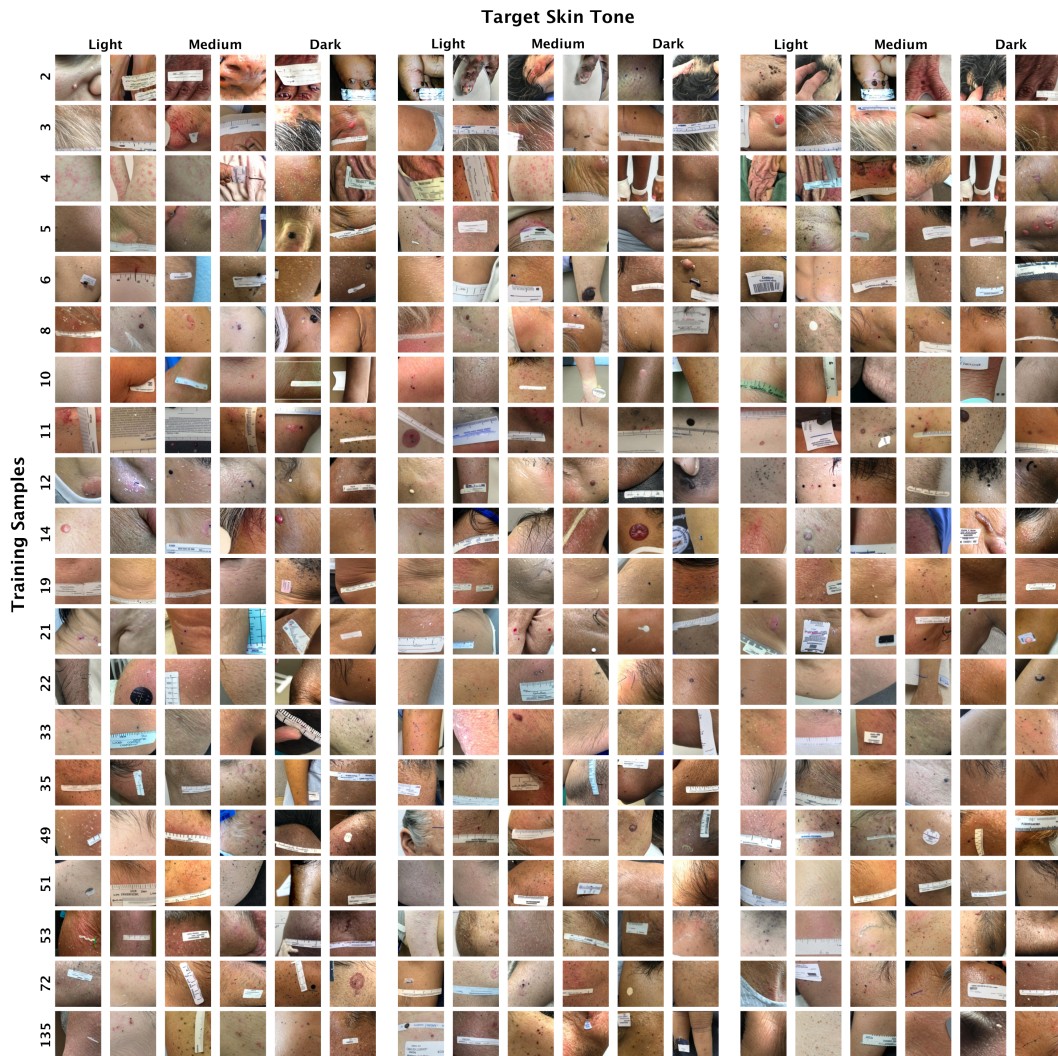

Figure 9: **Data Efficiency Ablation**. We randomly sample three subsets of semantic synthetics equally between target skin tones. We observe increasing quality with increasing training data noting better human anatomy, increasingly realistic rulers, lesion quality and instruction following.

observation of random generative subsets. Figure 9 shows generation results with varying numbers of training samples.

Our findings indicate that generation quality improves significantly up to approximately 10 samples, after which gains become more incremental. We specifically note improvement in human anatomy, valid clinical rulers, lesion morphology (with the isolation of the target disease, instead of generating multiple lesions), and instruction following. Thus we determine our approach is effective with as few as 10 real training samples per disease class.

## H REVIEW PROTOCOL AND DISCARD CRITERIA

To evaluate the quality of our synthetic images, we establish a review protocol. The review was conducted by the paper authors who have previous experience in dermatological AI research.

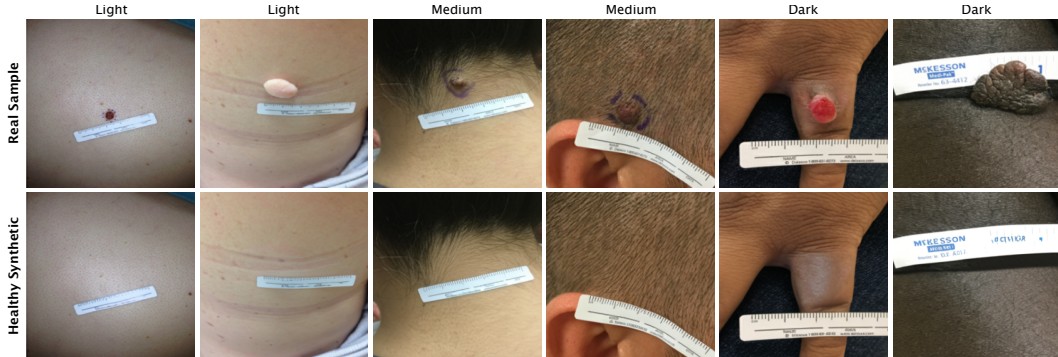

Figure 10: **Accepted Healthy Synthetics**. Further visualization of samples who passed our review protocol.

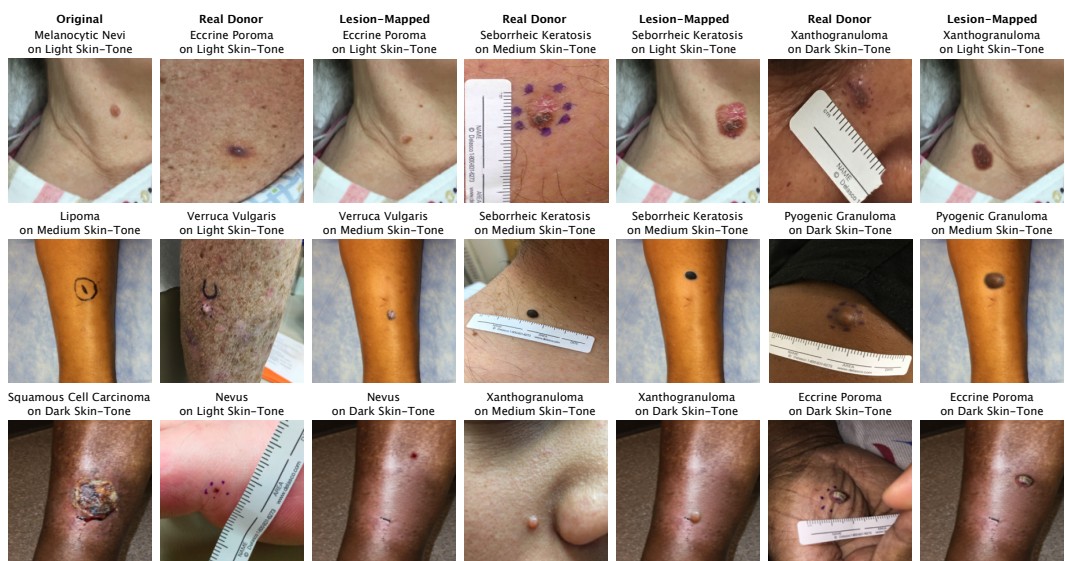

Figure 11: **Accepted Lesion-Mapped Synthetics**. Further visualization of samples that passed algorithmic constraints.

## H.1 HEALTHY SYNTHETICS

As there are a maximum possible 334 synthetics for this type, it was viable to manually review the full-set of images.

The discard criteria is as follows:

1. Original lesion remains present.

2. Unrealistic skin texture.

3. Incorrect skin tone.

4. Generative artifacts (unexpected holes, eyes, lighting, etc).

After review, we keep 309 healthy synthetics, discarding 25 (or 7%) of possible samples. Some illustrations of images that would trigger a discard appear on Figure 8 with accepted samples shown on Figure 10.

## H.2    Lesion-Mapped Synthetics

Since we generate 80,427 lesion-mapped synthetics, a manual review of the full set was not possible. We did however, algorithmically eliminate any output images that did not meet the padding criteria of 10 pixels, which should avoid strange placements and lighting, as previously mentioned. Given 309 synthetic healthy images, 334 sDDI masks and a single iteration loop, following the description given on Section 3.3, it is possible to generate 103,206 synthetics. However, the algorithmic constraint discards 22,779 samples (or 22%) of possible samples. We show additional lesion-mapped samples, along with their real donors, on Figure 11.

## H.3    Semantic Synthetics

As we generate 185,400 semantic synthetics, a manual review of the full set was not possible. However, we ablate generation quality for random subsets of data along with their corresponding amount of training data. We observe that at around 10 training samples, image quality becomes viable. This is visualized on Figure 9.

Some observations of synthetics with fewer than 10 training samples include:

1. Unrealistic skin texture.
2. Anatomical inconsistencies.
3. Unrealistic or unnatural patterns in lesion appearance or placement.
4. Ruler or other generative artifacts.
5. Poor instruction following.

Given this criteria, do not include semantic synthetics produced with $\leq 10$ training samples on our classification experiments. This means that out of 185,400 samples, we include 60,255 (or about 33%) in the classification experiments. We do however publish all semantic synthetics regardless of their original number of training samples for further research and evaluation.

# I    Classification Experiments

This section provides detailed information about our classification experiments, including model architecture, training hyperparameters, data splitting strategy, fairness metrics computation, and statistical analysis.

## I.1    Model Architecture and Training Hyperparameters

For our classification experiments, we followed the contrastive disentanglement approach from (Du et al., 2023) with the patch-alignment improvements in (Aayushman et al., 2024). The classifier architecture consists of a Vision Transformer (ViT-B/16) pre-trained on ImageNet-21k.

In terms of hyperparamters, we follow (Aayushman et al., 2024). Adam optimizer with learning rate $1e^{-4}$, with a linear decay scheduler at step size 2 and a decay factor of 0.8. Batch size is set to 32 and train for a total of 20 epochs.

For Experiment 2 (fine-tuning on real data), we used a lower learning rate of $1e^{-5}$ and trained for an additional 20 epochs with early stopping.

The baseline model (referred to as "Baseline" in Table 2) uses a pre-trained ResNet-18 as the feature extractor and only cross-entropy loss on the final labels for optimization, following the approach described in (Aayushman et al., 2024). We compare our method with this baseline as well as with FairDisCo (Du et al., 2023).

## I.2    Data Splits and Leakage Prevention

To ensure valid evaluation and prevent data leakage, we carefully designed our experimental setup. For all experiments, we followed the same 80/20 train/test split ratio as used in previous work (Du

et al., 2023; Aayushman et al., 2024) when evaluating on the DDI dataset, enabling direct comparison with baseline methods.

For Experiment 1 (training exclusively on synthetic data), we utilized synthetic images from our cgDDI dataset, ensuring no overlap with test images from the real DDI dataset. This is accomplished by removing any lesion-mapped synthetics which are generated from image donors present on the test set (as split by the (Aayushman et al., 2024) seeds). Furthermore, we remove any semantic synthetic which was generated with target disease and healthy image prompt which together reconstruct an image which is present on the test set.

For Experiment 2 (fine-tuning on real data), we first trained the model on our synthetic cgDDI dataset, then fine-tuned it on the real DDI training set. The model was evaluated on the same real DDI test set as in Experiment 1 and as (Aayushman et al., 2024).

## I.3 STATISTICAL SIGNIFICANCE

To assess the statistical robustness of our results, and to directly compare against (Aayushman et al., 2024) we performed five experiments with differnet train/test split seeds. These are taken directly from (Aayushman et al., 2024) code repository and are ['S36' 'S37', 'S38', 'S39', 'S40'] which we believe to be seeds 36, 37, 38, 39 and 40. We calculated mean performance and standard deviation for each metric. The standard deviations reported in Table 2 reflect the variation across these splits.

## J  ETHICS, BROADER IMPACTS AND SAFEGUARDS

This section discusses ethical considerations, broader societal impacts, and safeguards related to our work.

### J.1  IRB APPROVAL AND DE-IDENTIFICATION PROTOCOL

This work builds exclusively on the publicly available DDI dataset (Daneshjou et al., 2022b), which was collected under Stanford IRB protocol numbers 36050 and 61146. All clinical images in the original dataset were de-identified following a strict protocol that removed all personally identifiable information (PII), including faces, tattoos, and unique identifiers.

No new human subjects were enrolled for this study, so no additional IRB approval was required. Our synthetic data generation further enhances privacy by creating artificial images that do not directly correspond to any real individual patient.

### J.2  TRANSFORMATIVE USE

To contextualize the transformative nature of our generation, we compare FID scores between simple augmentations and cgDDI methods against base DDI:

Table 13: FID scores comparing augmentation methods. cgDDI produces substantially different distributions than simple modifications.

| Method | FID ↓ |
|---|---|
| Duplication | 0.0 |
| Color Edits | 3.8 |
| Contrast | 7.3 |
| Rotation | 17.3 |
| Random Cropping | 23.0 |
| Gaussian Noise | 32.1 |
| cgDDI Lesion-mapped | 79.7 |
| cgDDI Healthy | 102.5 |
| cgDDI Semantic | 130.5 |

The significantly larger FID scores ($\geq$ 79.7) demonstrate that cgDDI creates substantially novel imagery rather than simple modifications, supporting the transformative nature of our contribution.

## J.3 BROADER IMPACTS DISCUSSION

We believe our cgDDI framework has several potential positive societal impacts:

1. Improved healthcare equity: By enabling the development of more fair and accurate dermatological AI systems, our work could help reduce healthcare disparities affecting underrepresented populations.

2. Enhanced medical education: Synthetic images can be used for medical education, providing dermatologists and other healthcare providers with exposure to diverse presentations of skin conditions.

3. Reduced privacy risks: Synthetic data can reduce the need for collecting and sharing sensitive patient images, mitigating privacy concerns.

However, potential negative impacts include:

1. Misuse of synthetic data: If not properly disclosed, synthetic images could be misrepresented as real clinical data.

2. Over-reliance on synthetic data: Exclusive training on synthetic data might not capture all the nuances present in real clinical scenarios.

3. Unrealistic expectations: Improvements in AI performance on synthetic data might not translate directly to real-world clinical settings without careful validation.

We believe negative risks can be mitigated with further improvement of our models via additional training and validation.

## J.4 SAFEGUARDS AND LICENSING

To mitigate potential risks and ensure responsible use of our work, we have implemented the following safeguards:

- Clear documentation: We provide comprehensive documentation about the synthetic nature of our dataset and its intended use.

- Data statements: Each synthetic image is accompanied by metadata clearly identifying it as synthetic and specifying the generation method.

- Open licensing: We release our code, models, and dataset under the Apache 2.0 license, enabling broad access with attribution.

- Ethical guidelines: We will include usage guidelines that encourage responsible application in research and educational contexts.

All resources are available at `https://anonymous.4open.science/r/ControllableGenDDI` with appropriate documentation and licensing information.

## K  LIMITATIONS

While our experiments demonstrate cgDDI improves classification performance and fairness while lowering dataset size limitations, extremely-efficient parametric training and generation (fewer than 10 samples) remains an open challenge in the vision community, and more so toward medical AI. We identify promising directions further reduction:

- **Few-shot adaptation**: Techniques like Model-Agnostic Meta-Learning (MAML) could potentially reduce requirements to 3-5 samples

- **Further prompt engineering**: Advanced prompt design might enable zero-shot generation for unseen diseases, such as explicitly describing in natural langue visual features associated with particularly rare morphologies

- **Cross-disease transfer**: Learning shared disease characteristics could enable better generalization

We leave systematic investigation of these approaches as future work, noting that our non-parametric lesion mapping already enables augmentation from single samples. Additionally, our approach provides generative control via prompting. However, instruction following is strongly influenced by sampling hyperparameters. Our parameter selection, while informed by preliminary experiments, was not exhaustively optimized due to computational costs. Finally, while we manually and algorithmically review generated samples, incorporating dermatologist review across all our subsets would measure dataset quality and provide information for filtering.

## L  REPRODUCIBILITY AND COMPUTE RESOURCES

This section provides detailed information about the computational resources and environment used for our experiments to facilitate reproducibility.

### L.1  HARDWARE AND GPU USAGE

All experiments were conducted using the following hardware:

- GPU: NVIDIA L4 24GB

- CPU: Intel Xeon @ 2.20GHz

- RAM: 53GB

- Storage: 112.6GB

The approximate compute time for each component was:

- Healthy synthetic generation: 2.5 GPU hours

- Lesion mapping: 30 CPU hours

- Textual inversion training: 45 GPU hours

- LoRA fine-tuning: 30 GPU hours

- Semantic synthetic generation: 250 GPU days

- Classification experiments: 10 GPU hours

The total compute resources used for this project amounted to approximately 6,000 GPU hours which we split across multiple compute instances.

### L.2  SOFTWARE ENVIRONMENT

Our implementation used the following software environment:

- Operating System: Ubuntu 22.04.4 LTS

- CUDA: 12.4

- Python: 3.11.12

In terms of packages, we leverage Google Colab pre-installed libraries. All additional dependencies and version information are documented in our code repository.

### L.3 CODE AND DATA ACCESS

All code, models, and datasets are publicly available at:

`https://anonymous.4open.science/r/ControllableGenDDI`

The repository includes:

- Source code for all components of the cgDDI framework
- Pre-trained model weights for textual inversion and LoRA adaptations
- Full synthetic dataset with metadata
- Documentation on usage and reproduction

