# OpenReview forum: "cgDDI: Controllable Generation of Diverse Dermatological Imagery for Fair and Efficient Malignancy Classification"
_ICLR.cc/2026/Conference — Submitted to ICLR 2026_

### Official Review · Reviewer_LEQ6 · 2025-10-21

**Soundness:** 2
**Presentation:** 3
**Contribution:** 2
**Rating:** 4
**Confidence:** 4

**Summary:**

This paper proposes cGDDI, a controllable generation framework designed to generate diverse dermatology images conditioned on key sensitive attributes. This model enables (1) generation of in-distribution healthy samples, (2) mapping lesions onto novel skin tones, and (3) efficient parametric generation from limited training samples.

Using cGDDI, the authors expand the DDI dataset by 400 x and compare models trained on synthetic data against prior fairness-focused models in dermatology image analysis.

**Strengths:**

* S1: This paper explores image generation for fairer dermatology image analysis, which is of interest to the community.

* S2: Using diffusion models for controllable generation on skin attributes is well-motivated. The design of three generative pipelines, including healthy synthetics, lesion-mapped synthetics, and semantic synthetics, is conceptually sound.

* S3: The paper is clear and easy to follow.

**Weaknesses:**

**Major Weakness**

* W1: The methodological novelty appears to be limited. The proposed framework resembles a straightforward adaptation of conditional diffusion models for skin image generation.

* W2: The experimental scope is narrow. The authors should compare with prior generative methods for dermatology, such as [R3] and the works mentioned in Tab. 1, as well as more skin-tone-fairness-focused methods, such as [R1] and [R2]. Even if they were not originally designed for the DDI dataset, reproducing or fine-tuning them on DDI (or evaluating cGDDI on external datasets like Fitzpatrick17k) would make the results more convincing.


[R1] Bayasi, Nourhan, et al. "BiasPruner: Mitigating bias transfer in continual learning for fair medical image analysis." Medical image analysis (2025).

[R2] Xu, Zikang, et al. "Fairadabn: Mitigating unfairness with adaptive batch normalization and its application to dermatological disease classification." MICCAI 2023.

[R3] Pakzad, Arezou, Kumar Abhishek, and Ghassan Hamarneh. "CIRCLe: Color invariant representation learning for unbiased classification of skin lesions." ECCVW 2022.

[R4] Naveed, Asim, et al. "Ra-net: Region-aware attention network for skin lesion segmentation." Cognitive Computation (2024).

**Questions:**

**Primary  Questions/Suggestions**

* QS1: Could the authors distinguish the proposed method with existing works? What is the methodological novelty of this paper?

 * QS2: Given the domain specificity, I feel this paper is more suitable for medical conferences such as MICCAI or IPMI. The paper's contribution is somewhat narrow for ICLR, which typically emphasizes methodological advances with broader applicability.

* QS3: Could the authors provide comparisons with SOTA models in skin tone fairness and generative approaches?

---

> ### Author Response · Authors · 2025-12-03
>
> We thank Reviewer LEQ6 for their constructive feedback and questions. We appreciate their acknowledgment that our paper "explores image generation for fairer dermatology image analysis, which **is of interest to the community**". Additionally, they commend that "using diffusion models for controllable generation on skin attributes **is well-motivated**" and note that "the design of three generative pipelines, including healthy synthetics, lesion-mapped synthetics, and semantic synthetics, **is conceptually sound**". Finally, they describe our paper as "**clear and easy to follow**".
>
> We now address the reviewer's concerns:
>
> **1. Methodological novelty**
>
> The reviewer would like us to emphasize the novelty of our contributions. As discussed above, we begin by noting that, as shown on Table 1, the majority of synthetic dermatology work does not publish their resulting dataset for community use or train exclusively on private data. Furthermore, no previous work to the best of knowledge, has collected healthy imaging from dermatologist-verified skin tone labels collected in a setting analogous to that of diseased samples for downstream use (lines 226-229). Additionally we found no generative work in the dermatological domain has leveraged Prior Preservation Loss during training which address two main issues when fine-tuning diffusion models: forgetting semantic knowledge (drift) and reduced output diversity, enabling more faithful generation in comparison to textual inversion (Zeng et al., 2024) alone (Section 3.4). Finally, we establish SOTA performance and fairness on the DDI benchmark and have now expanded our results to cover inter and intra-dataset validation with Fitzpatrick17k data (F17k).
>
> **2. Experimental scope**
>
> The reviewer would like to see additional evaluations and comparisons to previous work, against and in addition to those presented in Table 1 and 2. We note, from our survey of previous work in Table 1, that only one method (Segers et al., 2023) has published its full synthetic dataset output. This makes a broad set of direct comparisons and evaluations difficult. We considered training our classifier on (Segers et al., 2023) synthetic data for comparison; however, they train and test on an unknown subset of DDI. Without knowing this information, we risk data leakage (i.e., (Segers et al., 2023) could have seen classification test images during generative training time). On their GitHub, in file paper_experiments/Experiments/ddi_validation.py, they write:
>
> ```
> # test_data = pd.DataFrame(metadata.groupby(['FST_bin']).apply(lambda x: x.sample(n=32, random_state=random_state)).reset_index(drop=True))
> test_data = pd.read_csv("../Metadata/ddi_heldout.csv")
> ```
>
> This suggests they originally planned to randomly sample 32 images per skin-tone using a random seed (similar to our strategy as well as FairDisCo (2023) and PatchAlign (2024) against which we compare), but then switched to using a pre-defined CSV file, which we are unable to locate to the best of our ability.
>
> In terms of the additional methods shared:
>
> **R1 BiasPruner** is a Continual Learning classification method which does not rely on bias annotations. The authors note FairDisCo, a previous method against which we compare, sets “an upper bound on the performance” adding “Despite not using bias, BiasPruner exhibits slightly lower but comparable performance to FairDisCo”. Training FairDisCo or its sequel PatchAlign on cgDDI data significantly improves it (Table 2).
>
> **R2 FairAdaBN** proposes adaptive batch normalization and loss rebalancing for sensitive subgroups. The authors do not evaluate their model on DDI. Comparing F17k performance between FairAdaBN and PatchAlign it appears that the second method is superior (FairAdaBN: 84.72% accuracy vs PatchAlign: 88.6%).
>
> **R3 CIRCLe** is a discriminative method with a generative component which alters the skin-tone of an input image. The authors unfortunately do not publish their generated synthetic imagery. Regenerating is computationally prohibitive for us.
>
> **R4 RA-Net** is a dermoscopic image segmentation algorithm. The reviewer does not provide commentary for this reference, but we appreciate them sharing this with us. We have implement SAMv3 as a generalist segmentation model for automated masking.
>
> **3. Venue**
>
> We note that recent medical dataset contributions have been submitted and accepted into ICLR such as FairSeg (ICLR 2024) and MedTrinity-25M (ICLR 2025). We believe our work is simultaneously a dataset and methodological contribution appropriate for this venue.
>
> **4. Comparisons**
>
> Comparing against many previous works is difficult as many do not publish train or generated data (and re-generating is often computationally expensive), as well as miss-alignment around metrics. We report additional results on F17k and encourage further comparison by fully releasing code, models and data.
>
> We thank the reviewer for their insightful comments and suggestions.

---

### Official Review · Reviewer_Uv6J · 2025-10-29

**Soundness:** 3
**Presentation:** 3
**Contribution:** 3
**Rating:** 6
**Confidence:** 3

**Summary:**

The paper proposes cgDDI, a pipeline for generating controllable dermatology images across different skin tones. The pipeline combines (i) latent-diffusion inpainting to create high-fidelity healthy skin canvases, (ii) a non-parametric lesion-mapping algorithm to transplant real lesions onto valid skin regions, and (iii) parametric text-conditioned diffusion (textual inversion + LoRA with prior-preservation) for disease/skin-tone–aware synthesis. Using DDI/sDDI as the base, the authors synthesize 266,136 images spanning three data types (healthy, lesion-mapped, and semantic).

**Strengths:**

- The work is well-motivated, targeting the non-trivial research problem of synthesizing dermatological images to address data scarcity, bias, and imbalance. The paper is built on a thorough investigation of related work, positioning its contributions to address a clearly defined unsolved problem.

- The pipeline is described clearly and is technically sound. In particular, first generating in-distribution healthy images and reusing them for Prior Preservation Loss is an excellent idea: it provides a targeted regularization set that mitigates catastrophic forgetting/model drift during fine-tuning on small, domain-specific data.

- The experiments are well-designed and demonstrate effectiveness with tangible improvements in both classification accuracy and fairness metrics. The paper also provides qualitative evidence that the method works as intended.

- The commitment to open-sourcing the synthesized images and models is a significant plus and will be a valuable contribution to the research community.

**Weaknesses:**

- The framework's (specifically the inpainting and lesion-mapping components) reliance on high-quality, human-annotated segmentation masks (from sDDI) is a limitation. Since such masks are often unavailable for other dermatological datasets, this dependency may limit the practical applicability and scalability of the method to new data sources.

- All experiments are centered on the DDI dataset. While this is a high-quality, biopsy-confirmed dataset, there is no external validation on other datasets (e.g., SCIN, Fitzpatrick17k). This leaves the cross-dataset robustness and generalizability of the generative models under-explored. While the authors note this as a limitation, it remains a key area for future validation to prove the method's effectiveness out-of-distribution.

**Questions:**

- Other recent works, such as Wang et al. (2024), use mask-free image-to-image translation to circumvent the need for segmentation masks. What do the authors see as the primary advantages of their mask-based approach (inpainting + lesion-mapping) compared to these mask-free methods?
- The current framework controls for disease and skin tone. Does the model learn any implicit priors about the anatomical feasibility of a condition (e.g., certain lesions appearing on specific body parts)? Or could the parametric model, for example, generate a lesion on an anatomically implausible location?
- If feasible, please include a small blinded dermatologist study/test to strengthen the clinical validity of the synthesized images of diseases. It seems like human evaluation is only involved in generating healthy imagery.

---

> ### Author Response · Authors · 2025-12-03
>
> We thank Reviewer Uv6J for their valuable comments, feedback and suggestions. We appreciate their noting that our work is "well-motivated**, targeting the **non-trivial research problem** of synthesizing dermatological images to address data scarcity, bias, and imbalance" and is "built on a **thorough investigation** of related work, positioning its contributions to **address a clearly defined unsolved problem**". Furthermore, they describe our pipeline as "described clearly" and "technically sound," highlighting that "first **generating in-distribution healthy images and reusing them for Prior Preservation Loss is an excellent idea**". They also praise our "well-designed" experiments that "demonstrate effectiveness with tangible improvements in both classification accuracy and fairness metrics". Finally, they recognize that "the commitment to **open-sourcing the synthesized images and models is a significant plus and will be a valuable contribution** to the research community".
>
> We now address the reviewer's concerns:
>
> **1. Segmentation masks**
>
> The reviewer is concerned that our method is limited by the availability of human-annotated segmentation masks. While we had mentioned that our framework should be widely compatible with out-of-the-box segmentation algorithms, we have now collected results to demonstrate this compatibility. We leverage the Segment Anything 3 (SAMv3) model (Carion et al.
> (2025)) to process Fitzpatrick17k (F17k) which does not include pre-made segmentation maps through our full framework. We observe high-quality lesion and skin masks with default use and simple prompting of the SAMv3 model, leading to validated healthy synthetics produced by our method which then serve toward the rest of the downstream pipeline. New visualizations, details, results and discussion for the SAMv3 results and F17k processing are available on the latest revision of our manuscript.
>
> **2. Cross-dataset robustness**
>
> The reviewer notes that our work was primarily based on the DDI dataset. Our original intent relying uniquely on DDI was to ensure quality from biopsy-verified base data and to exhibit significant sample/label efficiency. However, we have expanded our work to covered an expert-verified subset of F17k, and we directly address cross-dataset robustness by mapping or generating lesions from one dataset to healthy samples in the other and by fine-tuning models which are then cross-verified on separate dataset test-sets. We find all stages of our method to be robust, showing similar generative quality to the DDI results. We also observe external validation of our fine-tuned checkpoints on the non-overlapping diseases between DDI and F17k. These results are shared in detail on the latest revision of our manuscript.
>
> **3. Pre-made mask advantages**
>
> The reviewer asks, compared to mask-free methods (such as Wang et al. (2024)), the advantages that our approach brings. The primary use of masking is to produce healthy samples, which then serve as lesion-mapping targets, generative prompts and for prior preservation loss which are all novel aspects of our work. While we observe these factors to be important in generative quality and output diversity, our latest round of SAMv3 results show that out-of-the-box segmentation algorithms can effectively serve the role of human-made masks, and while not as good as pixel-perfect masks made by experts (leading to a higher discard rate), we find the results to be analogous to those that do leverage pre-made masks. This is again show on the latest revision.
>
> **4. Anatomical priors**
>
> The reviewer wonders if our models learn implicit priors around anatomical feasibility of certain conditions. This is a very interesting research question, and in-theory, the models should capture the distribution of anatomical locations for which certain lesions might present. This is however, in our belief, contingent on training on a large collection of observations covering this distribution which is an inherent limitation of Dermatological AI, especially for rare diseases. However, our framework is controllable via textual or image prompts which can have different weights associated with them, perhaps if dermatologists have extensive experience noting a particular disease only appears in a certain region, they could prompt the system to sample that disease on that body part. We leave the further study of this phenomena for future work.
>
> **5. Dermatologist study**
>
> The reviewer would like us to conduct a blind test with a dermatologist to validate our synthetics. We note the paper authors who have years of experience in Dermatological AI research did fully validate healthy synthetics and subsets of lesion-mapped and semantic synthetics, leading to the discard criteria noted on the manuscript appendix. However, we are not board-certified dermatologists and would indeed benefit from this sort of review, as noted on the paper limitations section.
>
> We thank Uv6J for their valuable review.

---

### Official Review · Reviewer_1omd · 2025-10-31

**Soundness:** 2
**Presentation:** 3
**Contribution:** 3
**Rating:** 4
**Confidence:** 3

**Summary:**

This paper proposes a framework named cgDDI to generate diverse dermatological imagery aimed at improving the fairness of skin disease classification. The main contribution lies in the cgDDI dataset, which is built upon this framework and enables fair classification across race, sex, and age groups.

**Strengths:**

1. The paper addresses an important problem — bias in generative models, particularly in the context of dermatological image synthesis. The proposed cgDDI framework allows for controllable and fair image generation, which is valuable for fairness research.

2. The generated dataset constitutes a meaningful contribution to the community. It has great potential to support fair and robust skin disease diagnosis, especially for rare conditions. As most existing generative models face limitations in producing reliable medical images, such curated datasets can substantially promote the adoption of AI in clinical dermatology.

3. The paper provides a relatively comprehensive presentation of the framework, data generation process, and classification results, demonstrating a solid understanding of both methodology and application.

**Weaknesses:**

1. The experimental results are somewhat unclear. In Table 2, the "Method" column refers to different combinations of datasets, which indeed suggests that synthetic data contributes to fairer classification. However, these findings are limited to the dataset proposed in this work. It would strengthen the claim to include comparisons with other datasets, as in Table 1.

2. The technical novelty appears limited. Most components seem to be adapted from existing methods, and the unique methodological contribution is not clearly articulated. The authors should clarify which elements of cgDDI are genuinely novel and how they differ from prior works.

3. Section 3 is difficult to follow in terms of logical flow and presentation. The methods are described mostly in text form, making it hard to identify the key design ideas. It would be helpful to include formulations, diagrams, or schematic illustrations to make the framework easier to understand.

**Questions:**

1. Is this paper more like a dataset contribution?

2. Whether the proposed framework can be applied to other base data, like the MIMIC dataset.

---

> ### Author Response · Authors · 2025-12-03
>
> We thank Reviewer 1omd for their valuable comments, feedback and suggestions. We appreciate their recognition that our work "**addresses an important problem** — bias in generative models" adding that the cgDDI framework "is **valuable for fairness research**". Additionally, they highlight that our dataset "constitutes a **meaningful contribution** to the community" with "great potential to support fair and robust skin disease diagnosis, especially for rare conditions" and note that our work can "**substantially promote the adoption** of AI in clinical dermatology". Finally, they commend our "**comprehensive presentation** of the framework, data generation process, and classification results, demonstrating a **solid understanding of both methodology and application**".
>
> We now address the reviewer's concerns:
>
> **1. Additional datasets**
>
> The reviewer shares their interest in the inclusion of additional data under the cgDDI framework, similar to some previous works described on Table 1 and further evaluation of how synthetic data, other than those originating from DDI, can improve performance in addition to the results on Table 2. Our original intent relying uniquely on DDI was to ensure quality from biopsy-verified base data and to exhibit significant sample/label efficiency.
>
> We are happy to share that, by processing a expert-verified subset of the Fitzpatrick17k (F17k) dataset through our framework, we are able to further validate our method on unseen disease, morphologies and imaging settings. Using this new data source, we:
> * Demonstrate off-the-shelf segmentation algorithms can effectively serve in-place of expert-made masks
> * Generate 46 additional healthy F17k synthetic samples
> * Lesion-map within F17k data a new collection of 1,124 samples
> * Learn textual-inversion tokens and fine-tune generative models for 8 new diseases covered by F17k, leading to 5,520 new semantic synthetics
> * Generate samples in a cross-dataset way, ie using healthy samples as generative prompts or mapping targets by moving lesions from one dataset to the other
> * Train classification models, showing a marked improvement in accuracy and fairness when leveraging synthetics both intra and inter-dataset.
>
> Visualizations, plots, tables and discussion on these new results have been added to the latest revision of our manuscript.
>
> **2. Technical novelty**
>
> We understand the reviewer would like us to emphasize the novel aspects of our contribution. We begin by noting that, as shown on Table 1, the majority of synthetic dermatology work does not publish their resulting dataset for community use or train exclusively on private data. Furthermore, no previous work to the best of knowledge, has collected healthy imaging from dermatologist-verified skin tone labels collected in a setting analogous to that of diseased samples for downstream use (lines 226-229). Additionally we found no generative work in the dermatological domain has leveraged Prior Preservation Loss during training which address two main issues when fine-tuning diffusion models: forgetting semantic knowledge (drift) and reduced output diversity, enabling more faithful generation in comparison to textual inversion (Zeng et al., 2024) alone (Section 3.4). Finally, we establish SOTA performance and fairness on the DDI benchmark.
>
> **3. Additional comments**
>
> We understand the reviewer would like to see additional formulations, diagrams and illustrations in Section 3. We will work to incorporate those in the next revision and by camera-ready. The reviewer asks if this paper is a dataset contribution, our opinion is that it is both a method and dataset paper relevant to the current needs and limitations present in the Dermatological AI field. Finally, the reviewer asks if the framework is compatible with other base data which we answer above.
>
> We hope this discussion is informative. We aim to address all reviewer concerns. Please do not hesitate to engage with us for further discussion. Once more, thank you for the valuable feedback.

---

### Author Response · Authors · 2025-12-03

# Response Summary

We understand this ICLR review period has been complicated and, while we would have appreciated the opportunity to engage in discussions, we thank all reviewers for their constructive feedback and the AC for conducting their meta-review. We revise our manuscript with new experiments and results placed at the top of the Appendix (pages 14-19) and will integrate them into the main body by camera-ready deadline as the page limit increases to 10. We summarize our responses here:

---

## 1. Algorithmic Lesion Masking

**Concern:** Reliance on human-annotated segmentation masks could limit scalability.

**Solution:** We demonstrate compatibility with SAMv3 (Segment Anything 3) for fully automated mask generation.

**Conclusion:** While expert masks yield lower discard rates, automated segmentation is fully compatible with our framework, enabling scalability to datasets without pre-made annotations.

**Results:** visualized on **Figure 5** with discussion in **Section A**.

---

## 2. Cross-Dataset Validation

**Concern:** Experimental scope is limited to DDI.

**Solution:** We process an expert-verified subset of F17k using our method and conduct generation and classification experiments.

**Conclusion:** We find high-quality synthesis and performance under this new data setting.

**Results:** found on **Sections A, B, C**. Generated images on **Figure 6**. We highlight the following:

### 2.2 F17k Classification Results

| Method | Accuracy | Light | Med | Dark | **PQD** |
|-|-|-|-|-|-|
| Baseline | 86.0% | 86.7% | 82.4% | 90.9% | 0.906 |
| Synth Only | 88.4% | 86.7% | **94.1%** | 81.8% | 0.869 |
| **Synth + Real** | **90.7%** | 86.7% | **94.1%** | 90.9% | **0.921** |

**Key Result:** +4.7% accuracy improvement and highest PQD achieved when combining our framework F17k synthetics with real data.

---

## 3. Cross-Dataset Transfer Experiments

**Concern:** Generalizability of our method.

**Solution:** We synthesize inter-dataset data between DDI and F17k and conduct cross-dataset classification experiments.

**Conclusion:** We find significant cross-dataset improvements.

We highlight the following:

### 3.1 Bidirectional Lesion Mapping

| Direction | Total Samples | Light | Medium | Dark |
|-|-|-|-|-|
| DDI lesions → F17k healthy | 13,822 | 6,593 | 4,267 | 2,962 |
| F17k lesions → DDI healthy | 9,394 | 3,103 | 3,272 | 3,019 |

Total 23,216 new lesion-mapped synthetics after discard criteria.

### 3.2 Cross-Dataset Semantic Generation

| Configuration | Total Images | Per Skin Tone |
|-|-|-|
| F17k diseases + DDI healthy prompts | 37,080 | 12,360 |
| DDI diseases + F17k healthy prompts | 8,970 | 2,990 |

Total 46,050 new lesion-mapped synthetics after discard criteria.

### 3.3 Cross-Dataset Classification (Zero-Shot Transfer)

Full results on **Table 8**, we highlight the following:

| Setting | Method | Accuracy |
|-|-|-|
| DDI → F17k | Baseline | 60.5% |
| DDI → F17k | **Synth Only** | **74.4%** |
| Mixed → DDI | Baseline | 83.2% |
| Mixed → DDI | Synth + Real | **86.7%** |
| Mixed → F17k | Baseline | 86.0% |
| Mixed → F17k | Synth + Real | **93.0%** |

**Key Result:** DDI-trained synthetics improve F17k accuracy by **+13.9%** despite minimal disease overlap (only 1 shared condition). Mixed dataset training improves performance by **+3.5-7%**.

---

## 4. Fairness Improvements

**EOM** (most important metric per Aayushman et al.): 69.6% → **86.6%** on DDI (+17%)
**PQD** leads in 5 of 6 cross-dataset experiments.

Additional discussion can be found on **Section C**.

---

## 5. Methodological in addition to dataset contribution highlights

1. **First method** to generate dermatologist-verified healthy skin imagery for downstream use (lesion-mapping, image prompting control and prior-preservation loss [PPL]).
2. **First use** of PPL in dermatological generation, address two main issues when fine-tuning diffusion models: forgetting semantic knowledge (drift) and reduced output diversity, enabling more faithful generation in comparison to textual inversion alone.
3. **Significant, diverse augmentation** >300× from as little as single sample observations, providing significant boost to downstream classifiers
4. **State-of-the-art** DDI performance: 90.9% (vs. 87.4% prior best) without changing classifier algorithm
5. **Open release** of 266k+ synthetic images, code, and models when most Medical AI papers retain training and generated data (Table 1).

---

We believe these comprehensive experiments directly address reviewer concerns. The new results demonstrate:
- Framework generalizability beyond DDI
- Practical applicability without expert masks
- Robust cross-dataset transfer
- Consistent fairness improvements
- Additional comments addressed in individual responses and in revised manuscript.

We are committed to incorporating all feedback, we welcome further discussion and we thank everyone involved in the review process.

---

### Public Comment · ~Héctor_Carrión1 · 2026-07-06
**Paper early-accepted to MICCAI 2026**

cgDDI has been early-accepted into MICCAI 2026, please find the latest work https://github.com/hectorcarrion/ControllableGenDDI

We thank the ICLR reviewers for helping us improve our work and suggesting the venue.

---

### Meta-Review · Area_Chair_66sq · 2026-01-06

**Summary:**

This paper aims to generate additional data to balance the training dataset. While this objective is practical and potentially useful, it limits the novelty of the technical contribution, as the method primarily focuses on data augmentation rather than introducing fundamentally new techniques. The approach is based on a conditional generative model to synthesize more samples, which is more like a dataset paper.

However, the data generation process is challenging because lesions are not always clearly segmented, which can cause the proposed model to fail. Furthermore, the experiments do not provide sufficient analysis regarding which generated data are useful for specific categories. Such evaluation should consider not only accuracy but also interpretation and reasoning, which are particularly important for medical applications.

**Reviewer Concerns:**

Additional experiments have been included to demonstrate the effectiveness of the approach.

The novelty of the proposed method is still limited. Moreover, the paper lacks intuitive results that could better illustrate and support the practical value of the method.

**Reviewer Scores:**

More experimental results will change reviewers' mind to improve the score, but the method innovation may not be satisfied for all reviewers.

---

### Decision · Program_Chairs · 2026-01-26

Reject